# MODELING MULTIMODAL ALEATORIC UNCERTAINTY IN SEGMENTATION WITH MIXTURE OF STOCHASTIC EXPERTS

**Zhitong Gao**[1], **Yucong Chen**[1], **Chuyu Zhang**[1,2], **Xuming He**[1,3]
[1]ShanghaiTech University, Shanghai, China    [2]Lingang Laboratory, Shanghai, China
[3]Shanghai Engineering Research Cente of Intelligent Vision and Imaging, Shanghai, China
`{gaozht,chenyc,zhangchy2,hexm}@shanghaitech.edu.cn`

## ABSTRACT

Equipping predicted segmentation with calibrated uncertainty is essential for safety-critical applications. In this work, we focus on capturing the data-inherent uncertainty (aka aleatoric uncertainty) in segmentation, typically when ambiguities exist in input images. Due to the high-dimensional output space and potential multiple modes in segmenting ambiguous images, it remains challenging to predict well-calibrated uncertainty for segmentation. To tackle this problem, we propose a novel mixture of stochastic experts (MoSE) model, where each expert network estimates a distinct mode of the aleatoric uncertainty and a gating network predicts the probabilities of an input image being segmented in those modes. This yields an efficient two-level uncertainty representation. To learn the model, we develop a Wasserstein-like loss that directly minimizes the distribution distance between the MoSE and ground truth annotations. The loss can easily integrate traditional segmentation quality measures and be efficiently optimized via constraint relaxation. We validate our method on the LIDC-IDRI dataset and a modified multimodal Cityscapes dataset. Results demonstrate that our method achieves the state-of-the-art or competitive performance on all metrics. [1]

## 1 INTRODUCTION

Semantic segmentation, a core task in computer vision, has made significant progress thanks to the powerful representations learned by deep neural networks. The majority of existing work focus on generating a single (or sometimes a fixed number of) segmentation output(s) to achieve high pixel-wise accuracy (Minaee et al., 2021). Such a prediction strategy, while useful in many scenarios, typically disregards the predictive uncertainty in the segmentation outputs, even when the input image may contain ambiguous regions. Equipping predicted segmentation with calibrated uncertainty, however, is essential for many safety-critical applications such as medical diagnostics and autonomous driving to prevent problematic low-confidence decisions (Amodei et al., 2016).

An important problem of modeling predictive uncertainty in semantic segmentation is to capture *aleatoric uncertainty*, which aims to predict multiple possible segmentation outcomes with calibrated probabilities when there exist ambiguities in input images (Monteiro et al., 2020). In lung nodule segmentation, for example, an ambiguous image can either be annotated with a large nodule mask or non-nodule with different probabilities (Armato III et al., 2011). Such a problem can be naturally formulated as label distribution learning (Geng & Ji, 2013), of which the goal is to estimate the conditional distribution of segmentation given input image. Nonetheless, due to the high-dimensional output space, typical multimodal characteristic of the distributions and limited annotations, it remains challenging to predict well-calibrated uncertainty for segmentation.

Most previous works (Kohl et al., 2018; 2019; Baumgartner et al., 2019; Hu et al., 2019) adopt the conditional variational autoencoder (cVAE) framework (Sohn et al., 2015) to learn the predictive distribution of segmentation outputs, which has a limited capability to capture the multimodal distribution due to the over-regularization from the Gaussian prior or posterior collapse (Razavi et al.,

---

[1]Code is available at `https://github.com/gaozhitong/MoSE-AUSeg`.

2018; Qiu & Lui, 2021). Recently, Monteiro et al. (2020) propose a multivariate Gaussian model in the logit space of pixel-wise classifiers. However, it has to use a low-rank covariance matrix for computational efficiency, which imposes a restriction on its modeling capacity. To alleviate this problem, Kassapis et al. (2021) employ the adversarial learning strategy to learn an implicit density model, which requires additional loss terms to improve training stability (Luc et al., 2016; Samson et al., 2019). Moreover, all those methods have to sample a large number of segmentation outputs to represent the predictive distribution, which can be inefficient in practice.

In this work, we aim to tackle the aforementioned limitations by explicitly modeling the multimodal characteristic of the segmentation distribution. To this end, we propose a novel *mixture of stochastic experts* (MoSE) (Masoudnia & Ebrahimpour, 2014) framework, in which each expert network encodes a distinct mode of aleatoric uncertainty in the dataset and its weight represents the probability of an input image being annotated by a segmentation sampled from that mode. This enables us to decompose the output distribution into two granularity levels and hence provides an efficient and more interpretable representation for the uncertainty. Moreover, we formulate the model learning as an *Optimal Transport* (OT) problem (Villani, 2009), and design a *Wasserstein*-like loss that directly minimizes the distribution distance between the MoSE and ground truth annotations.

Specifically, our MoSE model comprises a set of stochastic networks and a gating network. Given an image, each expert computes a semantic representation via a shared segmentation network, which is fused with a latent Gaussian variable to generate segmentation samples of an individual mode. The gating network takes a semantic feature embedding of the image and predicts the mode probabilities. As such, the output distribution can be efficiently represented by a set of samples from the experts and their corresponding probabilities. To learn the model, we relax the original OT formulation and develop an efficient bi-level optimization procedure to minimize the loss.

We validate our method on the LIDC-IDRI dataset (Armato III et al., 2011) and a modified multimodal Cityscapes dataset (Cordts et al., 2016; Kohl et al., 2018). Results demonstrate that our method achieves the state-of-the-art or competitive performance on all metrics.

To summarize, our main contribution is three-folds: (i) We propose a novel mixture of stochastic experts model to capture aleatoric uncertainty in segmentation; (ii) We develop a Wasserstein-like loss and constraint relaxation for efficient learning of uncertainty; (iii) Our method achieves the state-of-the-art or competitive performance on two challenging segmentation benchmarks.

## 2 RELATED WORK

**Aleatoric uncertainty in semantic segmentation**  Aleatoric uncertainty refers to the uncertainty inherent in the observations and can not be explained away with more data (Hüllermeier & Waegeman, 2021). Early works mainly focus on the classification task, modeling such uncertainty by exploiting Dirichlet prior (Malinin & Gales, 2018; 2019) or post-training with calibrated loss (Sensoy et al., 2018; Guo et al., 2017; Kull et al., 2019). In semantic segmentation, due to the high-dimensional structured output space, aleatoric uncertainty estimation typically employs two kinds of strategies. The first kind (Kendall & Gal, 2017; Wang et al., 2019; Ji et al., 2021) ignore the structure information and may suffer from inconsistent estimation. In the second line of research, Kohl et al. (2018) first propose to learn a conditional generative model via the cVAE framework (Sohn et al., 2015; Kingma & Welling, 2014), which can produce a joint distribution of the structured outputs. It is then improved by Baumgartner et al. (2019); Kohl et al. (2019) with hierarchical latent code injection. Other progresses based on the cVAE framework attempt to introduce an additional calibration loss on the sample diversity (Hu et al., 2019), change the latent code to discrete (Qiu & Lui, 2021), or augment the posterior density with Normalizing Flows (Valiuddin et al., 2021). Recently, Monteiro et al. (2020) directly model the joint distribution among pixels in the logit space with a multivariate Gaussian distribution with low-rank parameterization, while Kassapis et al. (2021) adopt a cascaded framework and an adversarial loss for model learning. Different from those methods, we adopt an explicit multimodal framework training with a Wasserstein-like loss, which enables us to capture the multimodal characteristic of the label space effectively.

**Optimal Transport and Wasserstein distance**  The optimal transport problem studies how to transfer the mass from one distribution to another in the most efficient way (Villani, 2009). The corresponding minimum distance is known as Wasserstein distance or Earth Mover's distance in

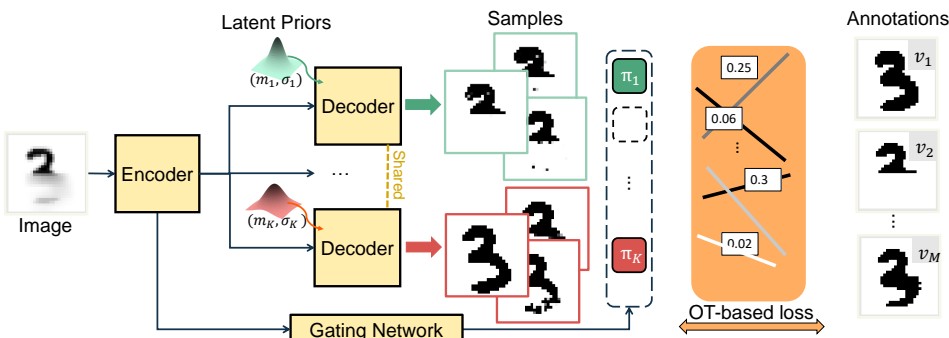

Figure 1: Given an ambiguous image as input, our model consists of several expert networks and a gating network and produces segmentation samples with corresponding weights. It represents the aleatoric uncertainty at two granularity levels, (e.g. the input image can be annotated as either "2" or "3" and also has variations near boundaries.) and is trained by an Optimal-Transport-based loss function which minimizes the distance between the MoSE outputs and ground truth annotations.

some special cases (Rüschendorf, 1985), which has attracted much attention in the machine learning community (Torres et al., 2021). In multi-class segmentation, Fidon et al. (2017; 2020) adopt the Wasserstein distance at the pixel level to inject prior knowledge of the inter-class relationships. We apply the Optimal Transport theory for calibrating the aleatoric uncertainty in segmentation, where a Wasserstein-like loss between the predicted distribution and ground truth label distribution is minimized. Unlike most work either dealing with fixed marginal probabilities (Damodaran et al., 2020) or fixed locations (Frogner et al., 2015) in the Optimal Transport problem, we use parameterized marginal probabilities and jointly optimize marginal probabilities and segmentation predictions. Furthermore, we design a novel constraint relaxation to efficiently compute and optimize the loss.

**Mixtures in deep generative models** There has been a series of work exploring mixture models in VAE (Dilokthanakul et al., 2016; Jiang et al., 2017) or GAN (Yu & Zhou, 2018; Ben-Yosef & Weinshall, 2018) framework. However, VAEs require learning a posterior network while GANs need to train a discriminator and often suffer from training instability. By contrast, our method does not require any additional network for training and utilize an OT-based loss that can be easily integrated with traditional segmentation loss and be efficiently optimized.

**Other multiple output problems** There have been many works in the literature on predicting *multiple structured outputs* (Guzman-Rivera et al., 2012; Rupprecht et al., 2017; Ilg et al., 2018). Unlike ours, those models are not necessarily calibrated and can only produce a limited number of predictions. Another similar task is *image-to-image translation* (Isola et al., 2017), which considers transforming images from one style to another and is addressed by learning a conditional generative model. However, these methods (Isola et al., 2017; Zhu et al., 2017; Esser et al., 2018) seek more diverse images and place less emphasis on calibration. In contrast, we aim to model a well-calibrated conditional distribution on segmentation that can generate varying segments during inference.

## 3 METHOD

In this section, we introduce our method for capturing the aleatoric uncertainty in semantic segmentation. Specifically, we first present the problem formulation in Sec. 3.1, then describe the design of our MoSE framework in Sec. 3.2, and our OT formulation of the model training in Sec. 3.3.

### 3.1 PROBLEM SETUP AND FORMULATION

We aim to build a flexible model for capturing the multimodal aleatoric uncertainty in semantic segmentation, where each input image could be annotated in multiple ways due to its ambiguity. To this end, we formulate it as the problem of estimating the probability distribution of segmentation outputs given an input image. Formally, we assume an annotated image dataset $\mathcal{D} = \{x_n, y_n^{(i)}, v_n^{(i)}\}_{n=1,\cdots,D}^{i=1,\cdots,M}$

is available for training, where $x_n \in \mathbb{R}^{H \times W \times 3}$ is an image and has $M (\geq 1)$ distinct segmentation annotations $y_n^{(1)}, \cdots, y_n^{(M)}$ with corresponding relative frequencies $v_n^{(1)}, \cdots, v_n^{(M)} \in [0, 1]$, where $\sum_i v_n^{(i)} = 1$. Each annotation $y_n^{(i)} \in \mathcal{Y} := \{1, \cdots, C\}^{H \times W}$, where $C$ is the number of classes and $H, W$ are the height and width of the image, respectively. Here $\{y_n^{(i)}, v_n^{(i)}\}$ represent random samples from the underlying label distribution of $x_n$.

To represent the aleatoric uncertainty, we introduce a conditional probabilistic model on segmentation, denoted as $\mu_\theta(y \mid x) \in P_\mathcal{Y}$, where $P_\mathcal{Y}$ is the space of segmentation distributions. The model learning problem corresponds to minimizing a loss function $\ell : P_\mathcal{Y} \times P_\mathcal{Y} \to \mathbb{R}$ between our model output $\mu_\theta(\cdot|x_n)$ and the empirical annotation distribution of the image $x_n$, denoted as $\nu_n$, over the dataset $\mathcal{D}$. Namely, our learning task can be written as,

$$\min_\theta \frac{1}{D} \sum_{n=1}^{D} \ell(\mu_\theta(y \mid x_n), \nu_n) \quad \text{where } \nu_n := \sum_{i=1}^{M} v_n^{(i)} \delta(y_n^{(i)}) \tag{1}$$

In this work, we focus on developing an effective and easy-to-interpret uncertainty representation for semantic segmentation. In particular, we propose a novel *multimodal representation* in our model $\mu_\theta(y \mid x)$ and a *Wasserstein*-like distribution distance as our loss $\ell$, which admits an efficient minimization strategy via partial constraint relaxation. Below we will present our model design and learning strategy in detail.

## 3.2 THE MIXTURE OF STOCHASTIC EXPERTS (MoSE) FRAMEWORK

In order to capture the multimodal structure of annotations, we develop a *mixture of stochastic experts* (MoSE) framework, which decomposes the output uncertainty into two granularity levels. At the fine-level, we employ an expert network to estimate a distinct mode of the aleatoric uncertainty in the dataset, which typically corresponds to certain local variations in segmentation. At the coarse-level, we use a gating network to predict the probabilities of an input image being segmented by those stochastic experts, which depicts the major variations in the segmentation outputs.

Specifically, we assume there exist $K$ stochastic experts $\{\mathcal{E}_k\}_{k=1}^K$, each of which defines a conditional distribution of the segmentation $y$ given an input image $x$, denoted as $P_k(y|x)$. Each expert $\mathcal{E}_k$ is associated with a weight $\pi_k(x)$ predicted by a gating network, which represents the probability of an input being annotated in the mode corresponding to $\mathcal{E}_k$. We instantiate $P_k(y|x)$ as an implicit distribution that maps an expert-specific prior distribution $P_k(z_k)$ (where $z_k$ is a latent variable) and the input $x$ to the output distribution via a deep network $F(x, z_k)$. As such, given the input $x$, we are able to generate a segmentation sample $s \in \mathcal{Y}$ as follows: we first sample an expert id $k$ from the categorical distribution $\mathcal{G}(\boldsymbol{\pi}(x))$ where $\boldsymbol{\pi}(x) = [\pi_1(x), \cdots, \pi_K(x)]$. Then we sample a latent code $z_k$ from $P_k(z_k)$ and compute the prediction $s$ by the network $F(\cdot)$:

$$s = F(x, z_k) \quad \text{where } k \sim \mathcal{G}(\boldsymbol{\pi}(x)), z_k \sim P_k(z_k). \tag{2}$$

An overview of our MoSE framework is illustrated in Figure 1. Below we introduce our network design in Sec. 3.2.1 and present an efficient output representation in Sec. 3.2.2.

### 3.2.1 NETWORK DESIGN

We adopt an encoder-decoder architecture for our MoSE model, which consists of an encoder network that extracts an image representation, $K$ expert-specific prior models and a decoder network that generates the segmentation outputs, as well as a gating module for predicting the expert probabilities. We detail the design of the experts and the gating module below.

**Expert Network** The stochastic expert network $\mathcal{E}_k$ comprises an encoder $f_e(\cdot)$, a decoder $f_d(\cdot)$ and latent prior models $\{P_k(z_k)\}$. We expect each expert to generate segmentation outputs for different images with a consistent uncertainty mode. To achieve this, we introduce a set of input-agnostic Gaussian distributions $\{P_k(z_k) = N(m_k, \sigma_k I), k = 1, \cdots, K\}$ as the expert-specific priors, where $m_k \in R^L$ and $I$ is an $L \times L$ identity matrix. Given an input image $x$, we use a Conv-Net as the encoder $f_e(x)$ to compute an image representation $u_g$, followed by a Deconv-Net as the

decoder $f_d(u_g, z_k)$ which takes the feature $u_g$ and a sampled noise $z_k$ to generate a segmentation output for the $k$-th expert. Concretely, assume the global representation $u_g \in R^{F_1 \times H' \times W'}$ has $F_1$ channels and a spatial size of $H' \times W'$. The decoder first generates a dense feature map $u_d \in R^{F_2 \times H \times W}$ and then fuses with the noise $z_k$ to produce the output. Here we adopt a similar strategy as in Kassapis et al. (2021), which projects $z_k$ to two vectors $\tilde{z}_k^b = w_1 z_k \in R^{F_2}$ and $\tilde{z}_k^w = w_2 z_k \in R^{F_2}$, and generates an expert-specific dense feature map as $\bar{u}_d^k[:, i, j] = (u_d[:, i, j] + \tilde{z}_k^b) \cdot \tilde{z}_k^w$, $(i, j) \in [1, H] \times [1, W]$. Finally, we use three $1 \times 1$ convolution layers to generate pixel-wise predictions. For notation clarity, we denote all the parameters in the expert networks as $\theta_s$.

**Gating Module** The gating network shares the encoder $f_e(\cdot)$ and uses the global representation $u_g$ to predict the probabilities of experts. Specifically, we first perform the average pooling on $u_g$ and pass the resulting vector $\bar{u}_g = \text{AvgPool}(u_g) \in R^{F_1}$ through a three-layer MLP to generate the probabilities $\boldsymbol{\pi}(x)$. We denote the parameters in the gating module as $\theta_\pi$.

We note that our MoSE framework is flexible, which can be integrated into most modern segmentation networks with a small amount of additional parameters. Moreover, it can be easily extended to other model variants where the gating module takes the input features from other convolution layers in the encoder or the latent noise is fused with other deconvolutional layers in the decoder (cf. C.5).

### 3.2.2 UNCERTAINTY REPRESENTATION

Due to the implicit distributions used in our MoSE framework, we can only generate samples from expert networks. In this work, we consider two types of nonparametric representations for the predictive uncertainty $\mu_\theta(y \mid x)$. The first representation utilizes the standard sampling strategy to generate a set of $N$ samples $\{s^{(i)}\}_{i=1}^N$ for an input image (cf. Eq.2), and approximates the uncertainty by the resulting empirical distribution $\frac{1}{N} \sum_{i=1}^N \delta(s^{(i)})$. The second method utilizes the MoSE framework and approximates the uncertainty distribution by a set of weighted samples. Specifically, we first generate $S$ samples from each expert, denoting as $\{s_k^{(i)}\}_{i=1}^S$ for the $k$-th expert, and then approximate the predictive distribution $\mu_\theta$ with the following $\hat{\mu}_\theta$:

$$\hat{\mu}_\theta(y \mid x) = \sum_{k=1}^K \frac{\pi_k(x)}{S} \sum_{i=1}^S \delta(s_k^{(i)}) \tag{3}$$

where $s_k^{(i)} = F(x, z_k^{(i)})$, and $z_k^{(i)} \sim N(m_k, \sigma_k I)$. It is worth noting that we can sample from multiple experts simultaneously as above, which leads to an efficient and compact representation of the segmentation uncertainty. For clarity, we denote the nonparametric representation of both forms as $\sum_{i=1}^N u^{(i)} \delta(s^{(i)})$ in the following, where $u^{(i)}$ is the sample weight and $N$ is the sample number.

### 3.3 OPTIMAL TRANSPORT-BASED CALIBRATION LOSS

The traditional segmentation loss functions typically encourage the model prediction to align with unique ground truth (e.g. Dice) or require an explicit likelihood function (e.g. Cross-Entropy). To fully capture the label distribution with our implicit model, we formulate the model learning as an *Optimal Transport* (OT) problem, which minimizes a *Wasserstein*-like loss between the model distribution $\mu_\theta(y \mid x_n)$ and the observed annotations $\nu_n$ of $x_n$ on the training set $\mathcal{D}$.

Specifically, our loss measures the best way to transport the mass of model distribution $\mu_\theta(y \mid x_n)$ to match that of the ground truth distribution $\nu_n$, with a given cost function $c : \mathcal{Y} \times \mathcal{Y} \to \mathbb{R}$. Based on this loss, our learning problem can be written as follows,

$$\min_\theta \sum_{n=1}^D W_c(\mu_\theta(y \mid x_n), \nu_n) \quad \text{where } W_c(\mu, \nu) := \inf_{\gamma \in \Pi(\mu, \nu)} \int_{\mathcal{Y} \times \mathcal{Y}} c(s, y) d\gamma(s, y). \tag{4}$$

Here $\Pi(\mu, \nu)$ denotes the set of joint distributions on $\mathcal{Y} \times \mathcal{Y}$ with $\mu$ and $\nu$ as marginals. Substituting in our discrete empirical distributions $\hat{\mu}_\theta(y \mid x) = \sum_{i=1}^N u^{(i)} \delta(s^{(i)})$ and $\nu = \sum_{j=1}^M v^{(j)} \delta(y^{(j)})$, we derive the following objective function:

$$\min_{\theta_s, \theta_\pi} \sum_n \sum_{i,j} \mathbf{P}_{ij}^* c(s_n^{(i)}(\theta_s), y_n^{(j)}) \quad \text{where } \mathbf{P}^* = \arg\min_{\mathbf{P} \in \mathbf{U}} \sum_{i,j} \mathbf{P}_{ij} \mathbf{C}_{ij}$$

$$\mathbf{U} := \{\mathbf{P} \in \mathbb{R}_+^{N \times M} : \mathbf{P} \mathbb{1}_M = \mathbf{u}_n(\theta_\pi), \mathbf{P}^T \mathbb{1}_N = \mathbf{v}_n\} \tag{5}$$

where $\mathbf{C} \in \mathbb{R}^{N \times M}$ is the ground cost matrix with each element $\mathbf{C}_{ij} := c(s^{(i)}, y^{(j)})$ being the pairwise cost. In our problem, we can use a commonly-adopted segmentation quality measure to compute the cost matrix. $\mathbf{P}$ is the coupling matrix that describes the transportation path, and $\mathbf{U}$ is the set for all possible values of $\mathbf{P}$ which satisfy the marginal constraints. This formulation can be regarded as a bi-level optimization problem, where the inner problem finds an optimal coupling matrix $\mathbf{P}$ and the outer problem optimizes the model output distribution under this coupling.

**Constraint Relaxation**    We note that the marginal probabilities, $\mathbf{u}(x; \theta_\pi)$, occurs in the constraint of the inner problem in Eq 5. As such, we can not directly use backpropagation to optimize the gating network parameters $\theta_\pi$ (Peyré et al., 2019). To tackle the challenge, we propose a novel imbalanced constraint relaxation. Our main idea is to have fewer constraints when optimizing the inner problem, and use the optimally solved matrix $\mathbf{P}$ as supervision for the gating network in the outer problem. Formally, we introduce the following relaxation:

$$\min_{\theta_s, \theta_\pi} \sum_n \sum_{i,j} \mathbf{P}_{ij}^* c(s_n^{(i)}(\theta_s), y_n^{(j)}) + \beta KL(\sum_j \mathbf{P}_{ij}^* || \mathbf{u}_n(\theta_\pi))$$
$$\text{where } \mathbf{P}^* = \arg\min_{\mathbf{P} \in \mathbf{U}'} \sum_{i,j} \mathbf{P}_{ij} \mathbf{C}_{ij} \quad \mathbf{U}' := \{\mathbf{P} \in \mathbb{R}_+^{N \times M} : \mathbf{P}\mathbb{1}_M \leq \gamma, \mathbf{P}^T\mathbb{1}_N = \mathbf{v}_n\}$$
(6)

where $\beta$ is used to balance the loss terms, and $\gamma \in [\frac{1}{N}, 1]$ is designed for preventing sub-optimal solutions in the early training stage. Empirically, we start with a value $\gamma_0 < 1$ and annealing it to 1 (meaning no constraint). It is worth noting that in the upper-level problem, the first term can be regarded as a normal segmentation loss for matched pairs of the predictions and ground truth labels, while the second term can be considered as optimizing the Cross-Entropy loss between the predicted probabilities and a pseudo label generated from the coupling matrix.

Our constraint relaxation has several interesting properties for model learning. Firstly, by moving the constraint of the predictive marginal distribution from the inner problem to the upper problem, we find it helps deal with the gradient bias (Bellemare et al., 2017; Fatras et al., 2020) when the ground truth annotations are limited for each image, which is common in real scenarios. In addition, our relaxed problem can be regarded as minimizing an approximation of the original problem, and achieves the same value when the KL term being 0. Finally, our relaxation leads to a more efficient alternating learning procedure. In particular, when $\gamma = 1$, we have only one constraint in the inner problem and this can be solved in linear time (Ignizio & Cavalier, 1994). We show the training procedure, details on solving the inner problem and theoretical analysis of Eq.6 in Appendix A, B.

## 4    EXPERIMENT

We evaluate our method on two public benchmarks, including LIDC-IDRI dataset (Armato III et al., 2011) and multimodal Cityscapes dataset (Cordts et al., 2016). Below we first introduce dataset information in Sec. 4.1 and experiment setup in Sec. 4.2 . Then we present our experimental results on the LIDC dataset in Sec. 4.3 and the Cityscapes dataset in Sec. 4.4.

### 4.1    DATASETS

**LIDC-IDRI Dataset**    The LIDC dataset (Armato III et al., 2011) consists of 1018 3D thorax CT scans, with each annotated by four anonymized radiologists.[2] For fair comparison, we use a pre-processed 2D dataset provided by Kohl et al. (2018) with 15096 slices each cropped to $128 \times 128$ patches and adopt the 60-20-20 dataset split manner same as Baumgartner et al. (2019); Monteiro et al. (2020). We conduct experiments in two settings, which train the model using either full labels or one randomly-sampled label per image as in the literature.

**Multimodal Cityscapes Dataset**    The Cityscapes dataset (Cordts et al., 2016) contains images of street scenes with 2975 training images and 500 validation images, labeled with 19 different

---

[2]A total of twelve radiologists participated in the image annotation and can revise their own annotations by reviewing the anonymized annotations of other radiologists. Therefore, the final disagreement among radiologists indicates true ambiguities and typically represents the aleatoric uncertainty (Armato III et al., 2011).

Table 1: Quantitative results on the LIDC dataset. The number of sampled outputs is shown in the parentheses aside the metrics. Our method achieves SOTA with smaller or comparable model size.

| Method | # label | GED ↓ (16) | GED ↓ (50) | GED ↓ (100) | M-IoU ↑ (16) | ECE ↓ (%) (16) | # param. |
|---|---|---|---|---|---|---|---|
| Kohl et al. (2018) | All | $0.320 \pm 0.030$ | - | $0.239 \pm$ N/A [†] | $0.500 \pm 0.030$ | - | 76.15M |
| Kohl et al. (2019) | | $0.270 \pm 0.010$ | - | - | $0.530 \pm 0.010$ | - | 87.51M |
| Hu et al. (2019) | | - | $0.267 \pm 0.012$ | - | - | - | 20.30M |
| Baumgartner et al. (2019) | | - | - | $0.224 \pm$ N/A | - | - | 74.82M |
| Monteiro et al. (2020) | | - | - | $0.225 \pm 0.002$ | - | - | 41.28M |
| Kassapis et al. (2021) | | $0.264 \pm 0.002$ | $0.248 \pm 0.004$ | $0.243 \pm 0.004$ | $0.592 \pm 0.005$ | $0.214$ [*] | 175.36M |
| **Ours - light** | | $0.219 \pm 0.002$ | $0.195 \pm 0.002$ | $0.190 \pm 0.003$ | $0.620 \pm 0.003$ | $0.070 \pm 0.013$ | 9.37M |
| **Ours** | | $\mathbf{0.218 \pm 0.003}$ | $\mathbf{0.195 \pm 0.002}$ | $\mathbf{0.189 \pm 0.002}$ | $\mathbf{0.624 \pm 0.004}$ | $\mathbf{0.064 \pm 0.015}$ | 41.60M |
| **Ours - compact** | | $\mathbf{0.195 \pm 0.005}$ | $\mathbf{0.187 \pm 0.003}$ | $\mathbf{0.186 \pm 0.002}$ | $\mathbf{0.635 \pm 0.003}$ | $\mathbf{0.054 \pm 0.015}$ | 41.60M |
| Kohl et al. (2018) | One | - | - | $0.445 \pm$ N/A [†] | - | - | 76.15M |
| Baumgartner et al. (2019) | | - | - | $0.323 \pm$ N/A | - | - | 74.82M |
| Monteiro et al. (2020) | | - | - | $0.365 \pm 0.005$ | - | - | 41.28M |
| **Ours** | | $\mathbf{0.252 \pm 0.004}$ | $\mathbf{0.229 \pm 0.005}$ | $\mathbf{0.223 \pm 0.005}$ | $\mathbf{0.596 \pm 0.003}$ | $\mathbf{0.105 \pm 0.009}$ | 41.60M |
| **Ours - compact** | | $\mathbf{0.228 \pm 0.004}$ | $\mathbf{0.221 \pm 0.004}$ | $\mathbf{0.220 \pm 0.005}$ | $\mathbf{0.605 \pm 0.003}$ | $\mathbf{0.090 \pm 0.011}$ | 41.60M |

[†] This score is taken from Baumgartner et al. (2019).

[*] This score is generated by running the pre-trained model provided by Kassapis et al. (2021).

semantic classes. To inject ambiguity, we follow Kohl et al. (2018) to randomly flip five original semantic classes into five new classes with certain probabilities. Specifically, the original classes of 'sidewalk', 'person', 'car', 'vegetation', 'road' are randomly transformed to sidewalk2', 'person2', 'car2', 'vegetation2', 'road2' with corresponding probabilities of 8/17, 7/17, 6/17, 5/17, 4/17. This brings a total of 24 semantic classes in the dataset, with $2^5 = 32$ modes of ground truth segmentation labels for each image (See Figure 3). For fair comparison, we follow the setting in (Kohl et al., 2018; Kassapis et al., 2021) and report results on the validation set.

## 4.2 EXPERIMENT SETUP

**Performance Measures** To measure how well the model captures the aleatoric uncertainty in segmentation, we follow Kohl et al. (2018; 2019); Baumgartner et al. (2019); Monteiro et al. (2020); Kassapis et al. (2021) to adopt two main metrics for evaluation: the intersection-over-union (IoU) based *Generalized Energy Distance* (GED) (Székely & Rizzo, 2013) and the *Hungarian-matched IoU* (Kuhn, 1955). We extend the latter to *Matched IoU* (M-IoU) by adopting network simplex algorithm (Cunningham, 1976) for the setting that the marginal distribution of predictions or ground truth is not uniform. Furthermore, we use the *expected calibration error* (ECE) (Guo et al., 2017) with 10 bins as a secondary metric to quantify the pixel-wise uncertainty calibration. For details on these metrics please refer to Appendix A.2. Below, we conduct all experiments in triplicate and report the results with mean and standard derivation.

**Implementation Details** We employ the same encoder-decoder architecture as in U-net (Ronneberger et al., 2015) and follow the model design as Baumgartner et al. (2019). On the LIDC dataset, we use $K = 4$ experts each with $S = 4$ samples. In the loss function, we use the IoU (Milletari et al., 2016) as the pair-wise cost function and set the hyperparameters $\gamma_0 = 1/2$, $\beta = 1$ for full-annotation case and $\beta = 10$ for one-annotation case. On the Cityscapes dataset, we use a slightly larger number $K = 35$ of experts each with $S = 2$ samples. We use the CE as the pair-wise cost function and set the hyperparameters $\gamma_0 = 1/32$, $\beta = 1$ for the loss. To stabilize the training, we adopt a gradient-smoothing trick in some cases and refer the reader to Appendix A.1 for details.

## 4.3 EXPERIMENTS ON THE LIDC-IDRI DATASET

We compare our method with the previous state of the arts and summarize the results under sample number N = 16, 50, 100 in the Table 1 and Appendix Table 5. We validate our method with two kinds of output representation (cf. Sec. 3.2.2): the standard-sampling-based representation denoted as 'Ours' and the weighted form as 'Ours - compact'. For the latter, we report the results by taking $S = 4$, 12, and 25 samples from each expert in order to generate the comparable number of total samples (i.e., 16, 48, and 100) as the standard-sampling form. We conduct experiments with a lighter backbone denoted as 'Ours - light' (cf. Appendix A) to make a fair comparison to Hu et al. (2019).

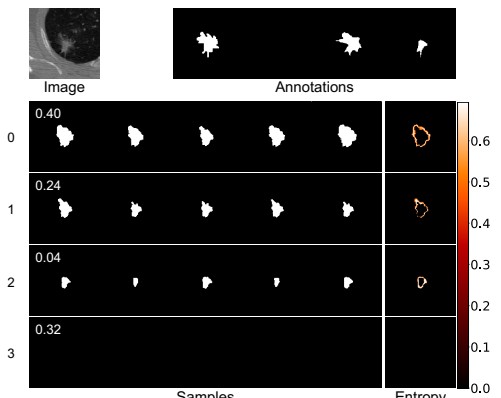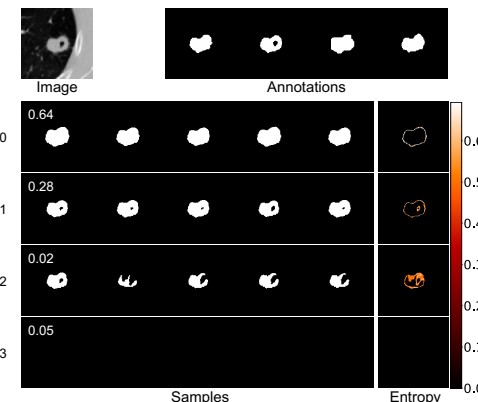

Figure 2: Qualitative results on the LIDC dataset. Each row of predictions represents a sampling sequence from an expert with weight denoted on the left head. The last line shows the entropy map.

Table 2: Ablation study on our model components, evaluated with sample number 16.

| Expert type | Expert weights | loss | GED ↓ | M-IoU ↑ | ECE ↓ (%) |
|---|---|---|---|---|---|
| stochastic | learnable / uniform | IoU loss | $0.533 \pm 0.001$ | $0.533 \pm 0.001$ | $0.277 \pm 0.017$ |
| stochastic | uniform | OT-based | $0.282 \pm 0.002$ | $0.545 \pm 0.007$ | $0.215 \pm 0.006$ |
| deterministic | learnable | OT-based | $0.246 \pm 0.006$ | $0.591 \pm 0.001$ | $0.142 \pm 0.003$ |
| stochastic | learnable | OT-based | $\mathbf{0.218 \pm 0.003}$ | $\mathbf{0.624 \pm 0.004}$ | $\mathbf{0.064 \pm 0.015}$ |

As shown in Table 1, our method is consistently superior to others with the standard-sampling form in all evaluation settings. Moreover, the compact weighted representation improves the performance further, especially in the small sample (16) case, and is more stable across different sample sizes. For the challenging one label setting, our method is less affected and also outperforms the others by a large margin, demonstrating its robustness to the number of annotations per image. We show the reliability plots in Appendix C.2. To validate that our MoSE model does capture the aleatoric uncertainty, we further conduct experiments with varying training data sizes in Appendix C.3.

Figure 2 visualizes our compact uncertainty representation on two cases. We can see that each expert predicts a distinct mode of aleatoric uncertainty with a shared style across images. The image-specific weighting of experts shows a coarse-level uncertainty, while the variations in samples from each expert demonstrate a fine-level uncertainty.

### 4.3.1 ABLATION STUDY

We conduct an ablation study of the model components on the LIDC dataset under the training setting with all (four) labels. We show the results with the sample number $S = 16$ under the standard-sampling form in Table 2, and leave the other settings to Appendix Table 6. In Row #1, we first analyze our OT-based loss by replacing it with the IoU loss (Milletari et al., 2016) calculated on every possible pair of predictions and labels. This leads to a performance drop of more than 0.3 in the GED score and the resulting model tends to predict an averaged label mask. Moreover, we modify our MoSE design by adopting a uniform weighting of experts in Row #2, and replacing the stochastic experts with deterministic ones in Row #3. For both cases, compared with our full model in the last row, we see a clear performance drop. Those comparison results validate the efficacy of our design. We also conduct more analysis experiments by varying different aspects of the model architecture. Please refer to Appendix C.4 and C.5 for details.

### 4.4 EXPERIMENTS ON THE MULTIMODAL CITYSCAPES DATASET

On this dataset, we compare our method with two previous works (Kohl et al., 2018; Kassapis et al., 2021). For fair comparison, we conduct two sets of experiments: the first follows Kassapis et al. (2021) to use predictions (Zoo, 2020) from a pretrained model as an additional input and use a lighter backbone, and the second uses no additional input as in Kohl et al. (2018). We evaluate our

Table 3: Quantitative results on the multimodal Cityscapes dataset.

| Method | Additional input | Sample number | GED ↓ | M-IoU ↑ | ECE ↓ (%) | # param. |
|---|---|---|---|---|---|---|
| Kohl et al. (2018) | ✗ | 16 | 0.206 ± N/A | 0.512 | 3.730 | 71.82M |
| Ours | | | **0.203 ± 0.002** | **0.580 ± 0.002** | **3.032 ± 0.018** | 41.63M |
| Kassapis et al. (2021) | ✓ | 16 | **0.164 ± 0.01** | 0.633 | 3.274 | 17.31M |
| Ours | | | 0.176 ± 0.002 | **0.638 ± 0.002** | 3.030 ± 0.021 | 41.63M |
| Ours - light | | | 0.177 ± 0.002 | **0.638 ± 0.001** | 2.973 ± 0.036 | 9.41M |
| Kassapis et al. (2021) | ✓ | 35 | **0.139** | 0.672 | 2.196 | 17.31M |
| Ours | | | 0.155 ± 0.002 | **0.690 ± 0.002** | 1.861 ± 0.015 | 41.63M |
| Ours - compact | | | 0.142 ± 0.001 | **0.761 ± 0.002** | 0.352 ± 0.025 | 41.63M |

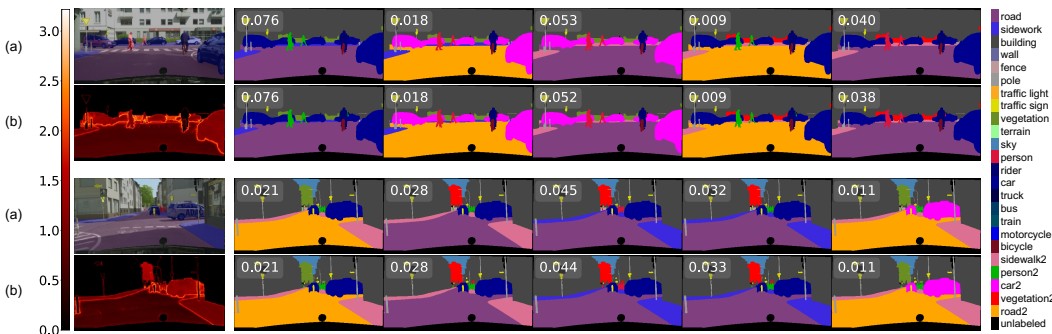

Figure 3: Qualitative results on the multimodal Cityscapes dataset. Within each case, (a) shows the image and a part of ground truth labels with corresponding probabilities, and (b) shows the entropy map, predictive samples and predicted probabilities.

model with the standard-sampling representation under 16 and 35 samples, as well as the compact representation under 35 samples (1 sample for each expert). As the previous works only provide their results on GED with 16 samples, we generate other results from the official pre-trained models and summarize the results in Table 3.

Overall, our method achieves the state-of-the-art M-IoU and ECE under both sample number settings, as well as comparable GED scores with Kassapis et al. (2021), demonstrating the superiority of our approach. The discrepancy in the GED performance is likely caused by the evaluation bias of the GED on sample diversity (Kohl et al., 2019), as Kassapis et al. (2021) typically produces more diverse outputs due to its adversarial loss. Moreover, we observe that our compact representation achieves better performance than the standard-sampling form under the same sample number (35), which demonstrates the effectiveness of such compact representation in multimodal uncertainty estimation. We show some visualization results in Figure 3. For clarity, we pair each ground truth label with the closest match in model predictions. We can see that our model can predict multimodal outputs with calibrated mode probabilities well. Furthermore, we include additional reliability plots, experimental analysis on the calibration of mode ratio, detailed ablation on model architecture, and more visualization results in the Appendix D.

## 5 CONCLUSION

In this work, we tackle the problem of modeling aleatoric uncertainty in semantic segmentation. We propose a novel mixture of stochastic experts model to capture the multi-modal segmentation distribution, which contains several expert networks and a gating network. This yields an efficient two-level uncertainty representation. To learn the model, we develop a Wasserstein-like loss that computes an optimal coupling matrix between the model outputs and the ground truth label distribution. Also, it can easily integrate traditional segmentation metrics and be efficiently optimized via constraint relaxation. We validate our method on the multi-grader LIDC-IDRI dataset and the muti-modal Cityscapes dataset. Results demonstrate that our method achieves the state-of-the-art or competitive performance on all metrics. We provide more detailed discussion on our method, including its limitations, in Appendix E.

## ACKNOWLEDGMENTS

This work was supported by Shanghai Science and Technology Program 21010502700 and Shanghai Frontiers Science Center of Human-centered Artificial Intelligence.

## REPRODUCIBILITY STATEMENT

The complete source code and trained models are publicly released at `https://github.com/gaozhitong/MoSE-AUSeg`. We provide the pseudo-code of our training procedure and explain the implementation details in Appendix A.1. We use the same segmentation backbone as Baumgartner et al. (2019); Monteiro et al. (2020) and provide details on the network architectures in Appendix A.1. We use two public datasets: LIDC and multimodal Cityscapes and follow the data processed procedure as previous work, as shown in Sec 4.1 and Appendix A.1.

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

## A  IMPLEMENTATION DETAILS

### A.1  TRAINING PROCEDURE

The overall training procedure is shown in algorithm 1. We use the weighted version of our non-parametric representation during training, as displayed in lines 3 - 9. Another option is to use standard-sampling representation (see Sec. 3.2) and utilize the Gumbel-Softmax trick (Jang et al., 2016) to enable backpropagation when drawing expert id $k$ from the categorical distribution $\mathcal{G}(\boldsymbol{\pi})$.

---

**Algorithm 1** Training Procedure

---

**Require:** Training data $\mathcal{D}$ with $M$ ground truth labels per image, number of experts $K$, number of
    samples per expert $S$, batch size $B$, leaning rate $\eta$, loss hyperparameters $\beta, \gamma$.
**Require:** initial expert network parameters $\theta_s$, initial gating network parameters $\theta_\pi$.
 1: **while** $\theta_s, \theta_\pi$ not converged **do**
 2:    Sample $\{x_n, y_n^{(j)}, v_n^{(j)}\}_{j=1,\cdots,M}^{n=1,\cdots,B}$ a batch from the training dataset $\mathcal{D}$.
 3:    **for** $k = 1, \cdots, K$ **do**
 4:        **for** $t = 1, \cdots, S$ **do**
 5:            Sample $\{z_n^{(k,t)}\}^{n=1,\cdots,B} \sim P_k$ a batch of expert prior codes.
 6:            $s_n^{(k,t)} \leftarrow F(x, z_n^{(k,t)}; \theta_s)$ with $n = 1, \cdots, B$.
 7:            $u_n^{(k,t)} \leftarrow \frac{1}{S}\pi_k(x; \theta_\pi)$ with $n = 1, \cdots, B$.
 8:        **end for**
 9:    **end for**
10:    Rearrange all outputs as $\{s_n^{(i)}, u_n^{(i)}\}_{i=1,\cdots,N}$, where $N = K \times S$.
11:    Compute pairwise cost matrix $\mathbf{C}_n$, with $\mathbf{C}_n[i,j] = c(s_n^{(i)}, y_n^{(j)})$, $n = 1, \cdots, B$.
12:    Solve the optimal coupling matrix $\mathbf{P}_n^* = \text{LP\_SOLVER}(\mathbf{C}_n, \mathbf{u}_n, \mathbf{v}_n, \gamma)$, $n = 1, \cdots, B$.
13:    $\ell \leftarrow \frac{1}{B} \sum_{n=1}^{B} (\langle \mathbf{P}_n^*, \mathbf{C}_n(\theta_s) \rangle + \beta KL(\mathbf{P}_n^* \mathbb{1}_M || \mathbf{u}_n(\theta_\pi)))$
14:    $\theta_s, \theta_\pi \leftarrow \text{Adam}(\nabla^{\text{soft}}\ell, \eta)$
15: **end while**

---

**Details on solving the coupling matrix**  In each iteration of training, we need to solve the best coupling matrix $\mathbf{P}^*$. This corresponds to the LP_SOLVER function module in algorithm 1  and is formally defined as:

$$\mathbf{P}^* = \operatorname*{arg\,min}_{\mathbf{P} \in \mathbf{U}'} \sum_{i,j} \mathbf{P}_{ij}\mathbf{C}_{ij} \quad \mathbf{U}' \coloneqq \{\mathbf{P} \in \mathbb{R}_+^{N \times M} : \mathbf{P}\mathbb{1}_M \leq \gamma, \mathbf{P}^T\mathbb{1}_N = \mathbf{v}_n\} \tag{7}$$

Especially, when $\gamma = 1$, the corresponding inequality constraint term vanished and $\mathbf{U}' \coloneqq \{\mathbf{P} \in \mathbb{R}_+^{N \times M} : \mathbf{P}^T\mathbb{1}_N = \mathbf{v}_n\}$. We can then solve the problem by moving the mass of each ground truth label to its nearest prediction. The time complexity is $O(N)$. When $\gamma < 1$, the problem can be solved by a standard linear programming solver like simplex or interior-point (Ignizio & Cavalier, 1994). We adopt a greedy strategy here, which empirically provides similar performance but with lower time complexity. Specifically, we first sort the ground truth labels according to their minimum cost to the predicted sample in ascending order. For each ground truth label, we also sort the predicted samples in terms of the corresponding cost in ascending order. Then, we go through the sorted ground truth labels and transfer their mass to the predicted samples in the sorted order given the constraint $\mathbf{P}\mathbb{1}_M \leq \gamma$ is satisfied. This algorithm has a time complexity of $O(NMlog(N))$.

**Details on computing Soft Gradient**  We observe that optimizing the loss (cf. Eq. 6) via a direct gradient descent could sometimes suffer from slow convergence and low performance due to large variance in the target signal $\sum_j \mathbf{P}_{ij}^*$ in the second term. This is partially caused by the random samples generated by the stochastic experts and unstable $\mathbf{P}^*$ inferred during the learning process. To address this, we adopt a simple smoothing technique to produce a stable gradient. Specifically, we group those experts with similar outputs during training, and average the gradients of the experts within each group. This prevents the gradients from changing dramatically in consecutive steps and resembles the Straight-Through method (Jang et al., 2016; Bengio et al., 2013).

We show the details for computing soft gradients in Algorithm 2 . Groups $g_1, \cdots, g_G$ are sets of sample indices, and can be computed by a clustering method. We here take a simple strategy and merge experts with similar outputs. Specifically, we initialize each group as an expert, and merge two experts $k, k'$ if $\text{IoU}(s^k, s^{k'}) \geq G_0$, where $s^k := \frac{1}{S} \sum_t s^{k,t}$ is the mean of samples for expert $k \in [K]$, $G_0$ is a hyper-parameter where we initialize as 1 and linearly decreased during training. Notice that we mainly apply this soft gradient operation on the Cityscapes dataset, where multiple experts generate similar predictions.

---

**Algorithm 2** Compute Soft Gradient

---

    **Input:** loss function $\ell$, model outputs $\{s^{(i)}, u^{(i)}\}_{i=1}^N$, coupling matrix $\mathbf{P}^*$, hyperparameter $G_0$.
    **Output:** soft gradients $\nabla_{\mathbf{u}}^{\text{soft}} \ell$.
1: **procedure** SOFTGRADIENT
2:     $g_1, \cdots, g_G \leftarrow \text{Clustering}(\mathbf{s}, G_0)$
3:     $\mathbf{o} \leftarrow \mathbf{P}^* \mathbb{1}_M$
4:     **for** $m \in 1 \cdots G$ **do**
5:         $\bar{u}^{(m)} \leftarrow \frac{1}{|g_m|} \sum_{i \in g_m} u^{(i)}$
6:         $\bar{o}^{(m)} \leftarrow \frac{1}{|g_m|} \sum_{i \in g_m} o^{(i)}$
7:         $\nabla_{u^{(i)}}^{\text{soft}} \ell(u^{(i)}, o^{(i)}) \leftarrow \nabla_{\bar{u}^{(m)}} \ell(\bar{u}^{(m)}, \bar{o}^{(m)}) \quad (i \in g_m)$
8:     **end for**
9:     **return** $\nabla_{\mathbf{u}}^{\text{soft}} \ell$
10: **end procedure**

---

**Details on Network Architechtures** For the backbone, we use the same encoder-decoder architecture as in U-net (Ronneberger et al., 2015) and follow the model design as (Baumgartner et al., 2019). Specifically, there are 6 down- and upsampling blocks, with each block consisting of three convolutional layers, each followed by a batch normalization layer and a ReLU-activation[3]. The global feature map $u_g \in R^{F_1 \times H' \times W'}$ has feature dimension $F_1 = 192$. The dense feature map $u_d \in R^{F_2 \times H \times W}$ has feature dimension $F_2 = 32$. The gating network contains three-layer MLP, with the first two layers having 192 neurons and the last layer having K neurons (K is the expert number). The three subsequent 1 x 1 convolution blocks after the decoder have 32 channels for the first two blocks and C channels for the last block ( C is the number of classes). In addition, in order to keep a fair comparison to the methods using a smaller backbone (Hu et al., 2019), we conduct experiments with lighter U-net backbone same as in (Hu et al., 2019) (decrease from the original 6 down-sample layer to 3 down-sample layer) [4] and keep other model design the same. We compare the model size of methods using the same segmentation backbone in Table 4

Table 4: Compare the model size of methods using the same segmentation backbone. Our model only adds a small number of parameters to the original Unet backbone. (* denotes the score is calculated according to the re-implementation by Baumgartner et al. (2019).)

| Method | #Backbone param. | #Additional param. (relative ratio) | #All param. |
|---|---|---|---|
| Kohl et al. (2018) | | 34.88M (84.5%) | 76.15M [*] |
| Baumgartner et al. (2019) | 41.27M | 33.55M (81.3%) | 74.82M |
| Monteiro et al. (2020) | | 0.01M (0.3%) | 41.28M |
| Ours | | 0.33M (0.8%) | 41.60M |
| Hu et al. (2019) | 9.05M | 11.32M (125.1%) | 20.37M |
| Ours - light | | 0.32M (3.5%) | 9.37M |

---

[3]Other details on backbone architecture can be referred to Baumgartner et al. (2019) and their code link `https://github.com/baumgach/PHiSeg-code`.

[4]Other details on the lighter backbone architecture can be referred to their code link `https://github.com/shihux/supervised_uq` and also the PyTorch implementation for the Probabilistic U-Net Kohl et al. (2018) model from `https://github.com/stefanknegt/Probabilistic-Unet-Pytorch`.

**Other Training Details** On the LIDC dataset, we apply random flip and random rotation augmentations same as Baumgartner et al. (2019); Monteiro et al. (2020). We use the latent code with dimension 1, which is empirically found to perform well and can be further refined. The models are trained for 200 epochs with batch size 16. On the Cityscapes dataset, we apply random horizontal flips, cropping, and scaling to size 128×128 same as Kassapis et al. (2021). We use a batch size of 12, latent code with dimension 5, and train the models for 1000 epochs. The ground truth marginal probabilities **v** (cf. 6) are initialized in uniform and annealing to the original values after some warm-up epochs. The additional prediction inputs are from the pretrained model xception71_dpc_cityscapes_trainfine provided by DeepLab Model Zoo [5]. As commonly practiced (Kassapis et al., 2021; Kohl et al., 2018), we ignore the unlabeled pixels when calculating loss. On both datasets, we use weight-decay with weight $1e^{-5}$ and Adam optimizer with an initial learning rate of $1e^{-3}$. The learning rates and loss slack factor $\gamma$ are linearly updated with scheduled updates. We do the model selection and tune hyperparameters according to the GED score on the validation set. We use two NVIDIA TITAN RTX GPUs on the LIDC dataset and four NVIDIA TITAN RTX GPUs on the Cityscapes dataset.

## A.2 EVALUATION METRICS

We aim to measure how well the model captures the multimodal aleatoric uncertainty in segmentation. The aleatoric uncertainty refers to the irreducible randomness in the data-generating process and can be represented by the ground truth label distribution (Monteiro et al., 2020; Hüllermeier & Waegeman, 2021). Following the literature (Kohl et al., 2018; 2019; Baumgartner et al., 2019; Monteiro et al., 2020; Kassapis et al., 2021), we use two distributional metrics: *generalized energy distance* (GED) and *matched-IoU* (M-IoU) to evaluate how the model fits the ground truth label distribution in the segmentation output space, which also demonstrates the quality of aleatoric uncertainty quantification. Due to the complexity of the high-dimensional segmentation distribution, both metrics use the sampling strategy to approximate the distributional distance via two sets of samples. In complementary to the sample-wise metrics, we also adopt the *expected calibration error* (ECE) (Guo et al., 2017) as an additional metric to measure pixel-wise uncertainty calibration. We note that while the ECE measures the quality of more general predictive uncertainty, our problem setting is dominated by the aleatoric uncertainty (cf. C.3 and D.3). Besides, as it is mainly used in the calibration of classification problems, the ECE can not measure the important multimodal and structural characteristics of segmentation uncertainty, and thus is adopted as a secondary metric. Below, we summarize the details of these three metrics. For clarity, we denote $\hat{y}$ as the hard prediction (after 'argmax' operation) to distinguish the soft prediction $s$ (after 'softmax' operation) . Other notations are kept consistent with previous content.

**Generalized energy distance (GED)** (Székely & Rizzo, 2013) measures the distance between two distributions by calculating the pair-wise distance $d(\cdot)$ = 1- IoU of samples from predictive distribution and samples from ground truth distribution. The expected value is then subtracted with the sample diversity of each distribution to get the final score:

$$\mathcal{D}^2_{GED}(\mu, \nu) := 2E_{\hat{y}\sim\mu, y\sim\nu}[d(\hat{y}, y)] - E_{y,y'\sim\nu}[d(y, y')] - E_{\hat{y},\hat{y}'\sim\mu}[d(\hat{y}, \hat{y}')]. \tag{8}$$

This metric is widely adopted by the previous literature (Kohl et al., 2018; 2019; Baumgartner et al., 2019; Monteiro et al., 2020; Kassapis et al., 2021). While effective in most situations, this metric potentially rewards sample diversity and sometimes gives a biased estimation of distance (Kohl et al., 2019).

**Matched-IoU (M-IoU)** (Kuhn, 1955; Cunningham, 1976) measures the distributional distance by solving an optimal matching matrix **P** between samples from the predictive distribution and ground truth distribution:

$$\mathcal{W}_{IoU}(\mu, \nu) := \max_{\mathbf{P}} \sum_{i,j} \mathbf{P}_{i,j} \text{IoU}(\hat{y}_i, y_j) \quad \text{s.t. } \mathbf{P}\mathbb{1}_M = \mathbf{u}, \mathbf{P}^T\mathbb{1}_N = \mathbf{v}. \tag{9}$$

It is previously adopted by Kohl et al. (2019); Kassapis et al. (2021) as compensation for GED measure since this metric does not favor sample diversity. We here use an extended version to enable a measure when one of the marginal distributions is not uniform. Specifically, on the LIDC

---

[5] Other details on this additional input can be referred to in Kassapis et al. (2021) and their public code: `https://github.com/EliasKassapis/CARSSS`.

dataset and evaluated with standard-sampling representation, we adopt the Hungarian matching algorithm (Kuhn, 1955) to solve matching matrix $\mathbf{P}$ same as Kohl et al. (2019); Kassapis et al. (2021). When evaluated with our compact representation, we adopt the network simplex algorithm (Cunningham, 1976) to solve $\mathbf{P}$. On the Cityscapes dataset where ground truth labels are aligned with non-uniform probabilities, we also adopt the network simplex algorithm.

**Expected calibration error (ECE)** (Guo et al., 2017) measures the difference between probability prediction and the real accuracy. It is defined as:

$$\text{ECE} := E_{\hat{P}}\left[\left|P(\hat{Y} = Y|\hat{P} = p) - p\right|\right] \tag{10}$$

Here, $Y$ is the random variable for the label, $\hat{Y}$ is the class prediction, and $\hat{P}$ is the associated probability. In our case, we get the pixel-wise label distribution and predictive distribution by marginalization and treat each pixel i.i.d. For the binary case in the LIDC dataset, we measure how the foreground class is calibrated. For the multi-class case in the Cityscapes dataset, we measure how the most-likely class is calibrated (aka confidence calibration) (Guo et al., 2017; Kull et al., 2019).

On the Cityscapes dataset, we follow the convention (Kohl et al., 2019; Kassapis et al., 2021) to only calculate metrics of the ten switchable classes.

## B    THEORETICAL ANALYSIS OF THE CONSTRAINT RELAXATION

We discuss the theoretical connection of our constraint relaxation to the original formulation in the following. For clarity, we consider a bi-level optimization formulation in which we adopt an alternating optimization process between $\theta_s$ (outer loop) and $\{P, \theta_\pi\}$ (inner loop). We first focus on the inner loop when the parameters for segmentation predictions $\theta_s$ is fixed, and analyze the optimization behavior of $\{P, \theta_\pi\}$ in the original form (Eq. 5) and relaxed form (Eq. 6). For simplicity, we rewrite them as following two functions:

$$l_1(u, P) = \langle P, C\rangle \quad u \in \mathbb{A}, P \in \mathbb{B}(u) = \{P \in \mathbb{R}^+ : P\mathbf{1}_M = u, P^T\mathbf{1}_N = v\}. \tag{11}$$

$$l_2(u, P) = \langle P, C\rangle + \beta KL(P\mathbf{1}_M||u) \quad u \in \mathbb{A}, P \in \mathbb{C} = \{P \in \mathbb{R}^+ : P^T\mathbf{1}_N = v\}. \tag{12}$$

where we use $C$ short for $C(s(\theta_s), y)$, and $u$ short for $u(\theta_\pi)$ for notation clarity. We use $\mathbb{A}$ to denote all possible value of $u$ that the gating network can generate, which also satisfies $u \in \mathbb{R}^+, \sum_i u_i = 1$. The inequality constraint in Eq. 6 is not included here in $l_2$ since it will eventually disappear due to annealing. We show in Proposition 1 that when $\beta$ is sufficiently large, $l_2$ achieves the same optimal solutions as $l_1$ when the KL term is zero. After that, when we alternates to optimizing $\theta_s$, the two problems are exactly the same as they use the same $P$ from the inner loop. Therefore the overall results of the original problem and the relaxed problem are the same.

**Proposition 1** *When $\beta$ is sufficiently large, $l_2$ achieves the same optimal solutions as $l_1$.*

*Proof:* We denote the domains for variables as $\Omega_1, \Omega_2$ in $l_1$ and $l_2$ correspondingly. It's obvious that $\Omega_1 \subseteq \Omega_2$, and we denote the complementary set $\Omega_c = \Omega_2 - \Omega_1$. We denote the optimal solutions as $P^*, u^* = \arg\min_{P,u \in \Omega_1} l_1, \hat{P}, \hat{u} = \arg\min_{P,u \in \Omega_2} l_2$. In the following, we consider two cases:

- When $P^*, u^* \in \Omega_1, \hat{P}, \hat{u} \in \Omega_1$, then $\hat{P}\mathbf{1}_M = \hat{u}$ is satisfied, and the KL term in $l_2$ is 0. Then we have $l_1(P^*, u^*) \leq l_1(\hat{P}, \hat{u}) = l_2(\hat{P}, \hat{u}) \leq l_2(P^*, u^*) = l_1(P^*, u^*)$. Therefore, the equality should hold and $l_1(\hat{P}, \hat{u}) = l_1(P^*, u^*) = \arg\min_{P,u \in \Omega_1} l_1(P, u)$, which means the optimal solution for $l_2$ is also the optimal solution for $l_1$.

- When $P^*, u^* \in \Omega_1, \hat{P}, \hat{u} \in \Omega_c$, then $KL(P\mathbf{1}_M||u) > 0$ must be true. Recall $\mathbb{A}$ is all possible values of $u$ that the gating network can generate, we consider two cases:

  (a) If $\hat{P}\mathbf{1}_M \in \mathbb{A}$, we can always find a solution $\hat{P}, u' \in \Omega_1$, by setting $u' = \hat{P}\mathbf{1}_M$. Then we have $l_2(P^*, u^*) \leq l_2(\hat{P}, u') < l_2(\hat{P}, \hat{u})$, which contradicts with the optimal of $\hat{P}, \hat{u}$.

  (b) If $\hat{P}\mathbf{1}_M \notin \mathbb{A}$, with a sufficient large $\beta$, we have $\langle\hat{P}, C\rangle + \beta KL(\hat{P}\mathbf{1}_M||\hat{u}) > \langle P^*, C\rangle + 0 = \langle P^*, C\rangle + \beta KL(P^*\mathbf{1}_M||u^*)$, which also contradicts with the optimal of $\hat{P}, \hat{u}$.

Therefore, when $\beta$ is sufficient large, the optimal solution for $l_2$ can only exist in $\Omega_1$, which is also the optimal solution for $l_1$.

In our optimization algorithm, we take a further step, alternating between P and u. Specifically, we first optimize $\bar{P} = \arg\min_{P \in \mathbb{C}} \langle P, C \rangle$ and then optimize $\bar{u} = \arg\min_{u \in \mathbb{A}} KL(P\mathbf{1}_M || u)$ (cf. Eq. 6). Compared to the joint optimization, this relaxed optimization strategy accelerating the optimization speed but have no guarantee to find the optimal solution as in the original form since the KL terms may not achieve zero. Specifically, we demonstrate in Proposition 2 that when the KL term can be optimized to 0, $\bar{P}, \bar{u}$ is still the optimal solution for $l_2$ and $l_1$. Empirically, we find that our model achieves KL divergence around 1e-5, which is much smaller than the scale of the segmentation loss in 1e-1 on the constructed multi-modal Cityscapes dataset.

**Proposition 2** *In our algorithm, when the KL term can be optimized to 0, $\bar{P} = \arg\min_{P \in \mathbb{C}} \langle P, C \rangle$, $\bar{u} = \arg\min_{u \in \mathbb{A}} KL(\bar{P}\mathbf{1}_M || u)$ is the optimal solution for $l_2$ and $l_1$.*

*Proof:* If $KL(\bar{P}\mathbf{1}_M || \bar{u}) = 0$, then $\bar{P}, \bar{u} \in \Omega_1$. Recall $\mathbb{B} = \{P \in \mathbb{R}^+ : P\mathbf{1}_M = u, P^T\mathbf{1}_N = v\}$ and $\mathbb{C} = \{P \in \mathbb{R}^+ : P^T\mathbf{1}_N = v\}$, since $\mathbb{B} \subseteq \mathbb{C}$, we have $\arg\min_{P \in \mathbb{C}} \langle P, C \rangle \leq \arg\min_{P \in \mathbb{B}} \langle P, C \rangle$, Therefore, $l_2(P^*, u^*) \leq l_2(\bar{P}, \bar{u}) = l_1(\bar{P}, \bar{u}) \leq l_1(P^*, u^*)$. Since $KL(P^*\mathbf{1}_M || u^*) = 0$, we have $l_2(P^*, u^*) = l_1(P^*, u^*)$, and hence the equality should hold, and $\bar{P}, \bar{u}$ is also an optimal solution for $l_1, l_2$.

## C    ADDITIONAL EXPERIMENTS ON LIDC

### C.1    COMPLEMENTARY RESULTS

In complementary to the results shown in the main text, we show the performance of our method with M-IoU and ECE under sample-number of 50, 100 in Table 5, and the complementary ablation study under the sample number of 50, 100 in Table 6. For short, we name each experiment by its replaced term and refer to the comprehensive explanation in Sec. 4.3.1.

Table 5: Other Quantitative results on LIDC-IDRI dataset. The ECE is in %.

| Method | # labels | M-IoU ↑(50) | M-IoU ↑(100) | ECE↓(50) | ECE↓(100) |
|---|---|---|---|---|---|
| Ours | All | $0.631 \pm 0.002$ | $0.636 \pm 0.002$ | $0.053 \pm 0.018$ | $0.052 \pm 0.017$ |
| Ours - compact | | $0.637 \pm 0.003$ | $0.638 \pm 0.002$ | $0.049 \pm 0.014$ | $0.049 \pm 0.014$ |
| Ours | One | $0.604 \pm 0.003$ | $0.606 \pm 0.003$ | $0.089 \pm 0.010$ | $0.085 \pm 0.012$ |
| Ours - compact | | $0.607 \pm 0.003$ | $0.607 \pm 0.003$ | $0.084 \pm 0.011$ | $0.083 \pm 0.012$ |

Table 6: Complementary results of the ablation study with sample number 50, 100. The ECE is in %.

| Method | GED ↓ (50) | GED ↓ (100) | M-IoU ↑ (50) | M-IoU ↑ (100) | ECE ↓ (50) | ECE ↓ (100) |
|---|---|---|---|---|---|---|
| IoU loss | $0.533 \pm 0.001$ | $0.533 \pm 0.001$ | $0.533 \pm 0.001$ | $0.533 \pm 0.001$ | $0.277 \pm 0.016$ | $0.277 \pm 0.016$ |
| Uniform expert weights | $0.256 \pm 0.003$ | $0.249 \pm 0.003$ | $0.554 \pm 0.006$ | $0.556 \pm 0.007$ | $0.209 \pm 0.008$ | $0.209 \pm 0.009$ |
| Deterministic Expert | $0.221 \pm 0.004$ | $0.216 \pm 0.004$ | $0.601 \pm 0.008$ | $0.603 \pm 0.009$ | $0.127 \pm 0.005$ | $0.121 \pm 0.003$ |
| Ours | $0.195 \pm 0.002$ | $0.189 \pm 0.002$ | $0.631 \pm 0.002$ | $0.636 \pm 0.002$ | $0.053 \pm 0.018$ | $0.052 \pm 0.017$ |

### C.2    RELIABILITY PLOTS

In Figure 4, we show the reliability plot of our model (left) and compare with the plot of Kassapis et al. (2021) (right). We generate the result by running the pre-trained model provided by Kassapis et al. (2021). For both methods, 16 samples are drawn for evaluation and we use the standard-sampling representation. This corresponds to the results on the last column of Table 1. We see that the confidence of our model calibrates well with the accuracy, resulting in a small gap in the curve and achieving a better pixel-wise calibration result than Kassapis et al. (2021).

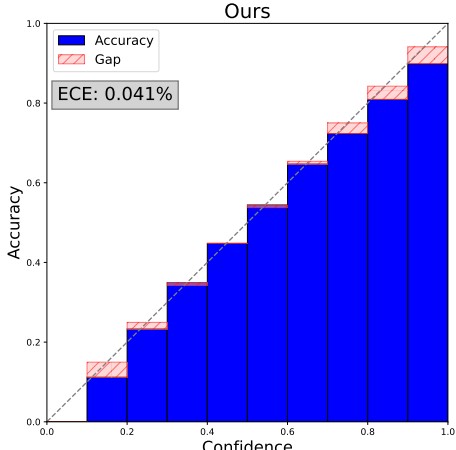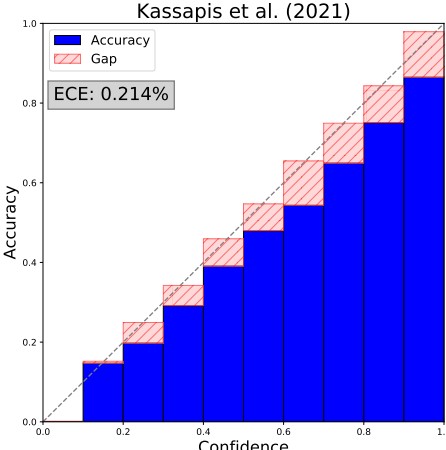

Figure 4: We show the reliability plot of our model (left) and compare with the plot of Kassapis et al. (2021) (right) on the LIDC dataset. For both methods, 16 samples are drawn for evaluation and we use the standard-sampling representation. We achieves 0.041% ECE (one of the three random runs in Table 1).

### C.3 ANALYSIS ON THE TYPE OF UNCERTAINTY

While specifically designed for modeling the data-inherent uncertainty (aleatoric), the outputs of our model may also contain the model uncertainty (epistemic). We note that our major evaluation metrics GED and M-IoU only measure how the model learns the aleatoric uncertainty. Therefore, the epistemic component in the output could lead to an additional error in these metrics. To explore how much the epistemic uncertainty accounts for the total predictive uncertainty in our model and how it affects the final results, we conduct the following experiments.

Specifically, we follow Kendall & Gal (2017) to utilize the property that epistemic uncertainty varies under different training data sizes while aleatoric part does not. By observing how the predictive uncertainty changes with different sizes of training data, we can infer how much the epistemic uncertainty accounts for in the predictive uncertainty. Besides, we also show how the metrics GED, M-IoU and ECE change, in order to analyze how the variation of epistemic uncertainty affects the final metrics. Experiments are done using all the labels and evaluated in 16 samples with standard-sampling representation.

As shown in Table 7, we change the size of the original LIDC training set to 1/2, 1/4 and summarize the uncertainty score by using pixel-wise entropy (same as Kendall & Gal (2017)) and sample diversity. The sample diversity is calculated as the sample-wise IoU distance, which is the same as the diversity term in calculating GED (cf. Eq.8). We find this score provides a better measure of the output variation under the class-imbalance situation. We see that both uncertainty scores increase slightly with the size of the training set decreasing. However, the magnitude of this change is in 1e-3, which is negligible when compared to the standard derivation of the repeated experiments. Therefore, we conclude that the predictive uncertainty contains only a small amount of epistemic uncertainty, which is nearly negligible. Besides, the changes in GED, M-IoU and ECE metrics are all in the same magnitude as the standard derivation of the repeated experiments and are at least one order of magnitude smaller than the difference in comparison results among methods in Table 1.

### C.4 ANALYSIS ON THE NUMBER OF EXPERTS

We analyze how expert number influences the model performance. To do this, we conduct experiments on the LIDC full label dataset under expert number = 1,2,4,6,8,10,15,20,30 where 4 is our default expert number that is used in other experiments. We show the GED score evaluated with 16 samples. As shown in Figure 5, when the expert number is smaller than 15, with the increasing

Table 7: We conduct experiments on the LIDC dataset with different training data sizes. Here, entropy and sample diversity are used to measure the uncertainty level of output in both pixel-wise and sample-wise. Experiments are done using all the labels and evaluated in 16 samples with standard-sampling representation.

| Train dataset | Entropy (%) | Sample Diversity | GED ↓ | M-IOU ↑ | ECE ↓ (%) |
|---|---|---|---|---|---|
| LIDC / 4 | $0.561 \pm 0.006$ | $0.509 \pm 0.010$ | $0.226 \pm 0.002$ | $0.615 \pm 0.006$ | $0.097 \pm 0.016$ |
| LIDC / 2 | $0.559 \pm 0.007$ | $0.505 \pm 0.011$ | $0.225 \pm 0.002$ | $0.619 \pm 0.004$ | $0.076 \pm 0.010$ |
| LIDC | $0.554 \pm 0.007$ | $0.501 \pm 0.007$ | $0.218 \pm 0.003$ | $0.624 \pm 0.004$ | $0.064 \pm 0.015$ |

number of experts, the model achieves a better GED score in general. However, when the expert number is much larger (eg. 20, 30 experts), the model performance decreases, corresponding to an increase of GED score. Intuitively, more experts correspond to a more powerful model capacity. Though with too many experts, there could be some duplicate ones predicting roughly similar predictions, which makes it difficult to learn a stable gating network. The results also demonstrate that the effectiveness of our method does not owe to a large number of experts, which is different from the working mechanism of ensembling model. By using 4 experts, we keep a balance between the model capacity and the expert compactness.

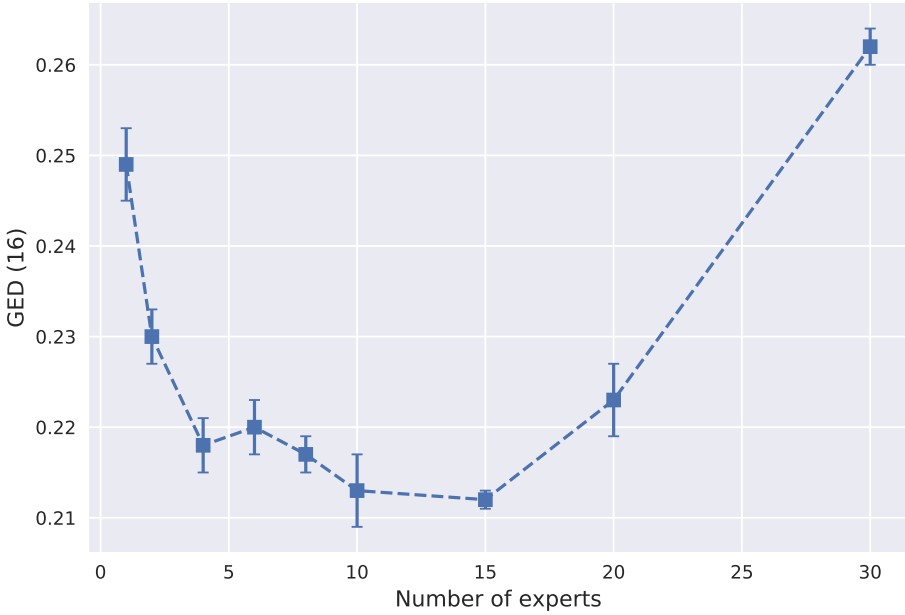

Figure 5: The GED score changes with the number of experts, evaluated with 16 samples in standard-sampling representation. Experiments are conducted on the LIDC dataset with the full label set.

## C.5 ANALYSIS OF THE MODEL ARCHITECTURE

Following, we explore the flexibility of our model design by conducting analysis experiments on three kinds of model variations: 1. different segmentation backbone; 2. the gating network takes input from different layers of the encoder; 3. the noise injects into the different layers of the decoder (cf. Sec. 3.2). Experiments are conducted on the LIDC dataset with full labels and evaluated with 16 samples in the standard-sampling representation. Results are shown in Table 8 and we summarize that with all variations explored, the model shows satisfactory results and consistently performs the

best compared with previous works (cf. Table 1), which demonstrates the flexibility of our MoSE design. Following, we analyze the experimental results for each variation in detail.

On row # 1, we change the currently used Unet architecture to DeepLabv3+ (Chen et al., 2018) with ResNet101[6]. The gating module takes feature from the ASPP output and noise injection are added in the last layer of the encoder. Other architecture designs and hyperparameters the kept same as default model. We see that, without further tuning, the experiment using DeepLabv3+ as the backbone has achieved satisfactory results on all metrics. This validates the feasibility of using other segmentation network as the backbone. On row # 2 and #3, we make the gating network take the input features from the 2nd and 4th layers from the overall 6-layer encoder. On row # 4 and #5, we conduct experiments with the latent noise fused with the 2nd and 4th layers from the overall 6-layer decoder. In comparison, we list the results of our default setting in the last row, which takes features from the last layer of the encoder and fuse latent noises in the last layer of the decoder. All other experimental settings are kept the same in these experiments. We observe that all these model variations show good performance on the final metrics, which validates the efficacy of using different levels of image features from the encoder network for the gating network input and injecting expert-specific noises in the different layers of decoder. For a new dataset, we suggest designing the gating network, its connection with the segmentation backbone, and noise injection way via prior knowledge or by architecture tuning.

Table 8: We analyze different model variations by replacing the backbone and changing the noise injection layer and the gating input layer. Experiments are done on the full-label LIDC dataset and evaluated with 16 samples in the standard-sampling representation. Below we denote L as the full layer of encoder and Q as the full layer of decoder. In our default Unet architecture, L = Q = 6.

| Backbone | Noise Injection Layer | Gating Input Layer | GED ↓ | M-IOU ↑ | ECE ↓ (%) |
|---|---|---|---|---|---|
| Deeplab v3+ | - | - | $0.237 \pm 0.005$ | $0.609 \pm 0.005$ | $0.104 \pm 0.018$ |
| Unet | 1/3 L | Q | $0.222 \pm 0.005$ | $0.620 \pm 0.004$ | $0.080 \pm 0.037$ |
| Unet | 2/3 L | Q | $0.218 \pm 0.001$ | $0.626 \pm 0.002$ | $0.082 \pm 0.009$ |
| Unet | L | 1/3 Q | $0.224 \pm 0.003$ | $0.625 \pm 0.001$ | $0.086 \pm 0.006$ |
| Unet | L | 2/3 Q | $0.220 \pm 0.001$ | $0.627 \pm 0.002$ | $0.082 \pm 0.011$ |
| Unet | L | Q | $0.218 \pm 0.003$ | $0.624 \pm 0.004$ | $0.064 \pm 0.015$ |

## C.6 ANALYSIS ON TRAINING STRATEGY

One difference between our method and previous works (Kohl et al., 2018; 2019; Baumgartner et al., 2019; Monteiro et al., 2020) is that we use all ground truth annotations per image in one gradient step during training. To analyze this impact, we conduct an ablation study where we random sample one ground truth label in each gradient step on the LIDC dataset and compare the results in Table 9. We observe that the performance of our method remains almost the same for such a setting, which indicates that our advantage does not come from using all groundtruth annotations in each gradient step.

Table 9: Analysis of our model when using only one ground truth label per gradient step. Experiments are conducted on the LIDC dataset with full annotations and evaluated with 16 samples.

| #gt per gradient step | GED ↓ | M-IOU ↑ | ECE ↓ (%) |
|---|---|---|---|
| One | $0.218 \pm 0.003$ | $0.627 \pm 0.005$ | $0.071 \pm 0.012$ |
| All (four) | $0.218 \pm 0.003$ | $0.624 \pm 0.004$ | $0.064 \pm 0.015$ |

## C.7 COMPARE WITH GMM-VAE MODELS

We also make empirical comparison between our method and VAEs with GMM prior (Dilokthanakul et al., 2016; Jiang et al., 2017; Lee et al., 2020). To do this, we replace the original gaussian prior distribution in Probablistic Unet (Kohl et al., 2018) by a GMM with K components, and approximates

---

[6]Details can be found in `https://github.com/jfzhang95/pytorch-deeplab-xception`.

the ELBO via Monte Carlo sampling as in Lee et al. (2020). Besides, we also evaluate a model variant by using input-agnostic Gaussian distributions and input-dependent categorical distribution as our model. We conduct experiments on the LIDC dataset, use K = 4 gaussian distirbutions (same as our expert number) and keep other settings the same as the Probabilistic Unet. The results are summarized in Table 10. We observe that the GMM prior do improves original Probabilistic Unet. However, our model still achieves the best performance, outperforming other models with a large margin (eg. 3% in GED). This demonstrates the efficacy of our design, especially our OT-based loss (for being one of the major difference).

Table 10: An empirical comparison of our method and VAEs with GMM prior. We use the reimplemented version of ProbUnet provided by Baumgartner et al. (2019) to keep a same Unet backbone.

| Method | Gaussian Prior(s) | GED ↓ | M-IOU ↑ | ECE ↓ (%) | # param. |
|---|---|---|---|---|---|
| ProbUnet | Input dependent | 0.298 ±0.010 | 0.527 ±0.007 | 0.118 ±0.012 | 76.15 |
| ProbUnet with GMM prior (1) | Input dependent | 0.254 ±0.008 | 0.597 ±0.005 | 0.111 ±0.013 | 76.18 |
| ProbUnet with GMM prior (2) | Input agnostic | 0.251 ±0.004 | 0.597 ±0.007 | 0.101 ±0.010 | 76.14 |
| Ours | Input agnostic | **0.218 ±0.003** | **0.624 ±0.004** | **0.064 ±0.015** | 41.60 |

## D  ADDITIONAL EXPERIMENTS ON CITYSCAPES

### D.1  RELIABILITY PLOTS

On the left of Figure 6, we show the reliability plot of our model with the sample number $S = 35$ under the compact representation. For comparison, we generate the reliability plot of the previous state of the art (Kassapis et al., 2021) by running the pre-trained model provided by Kassapis et al. (2021) with 35 samples and show the result on the right. We see that our model achieves a better calibration result.[7]

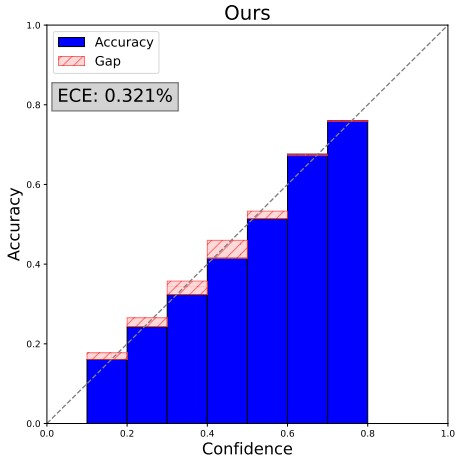
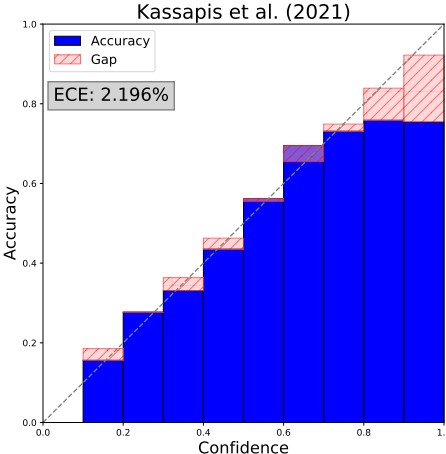

Figure 6: We show the reliability plots of our model (left) and Kassapis et al. (2021) (right) on the Cityscapes dataset evaluated with 35 samples. Our result is taken from one of the three random runs in Table 3 last column, last row.

### D.2  CALIBRATION IN MODE RATIO AND PIXEL FLIP RATIO

To have a further analysis of the predictive distribution and see how well it learns the characteristic of the multimodal ground truth distribution, we follow Kohl et al. (2018); Kassapis et al. (2021) to

---

[7]Notice that our model has all confidence scores less than 0.8. This is reasonable since the maximum occurrence probability of the evaluated ground truth class is $(1 - \frac{4}{17}) \approx 0.76$ (cf. Sec. 4.1).

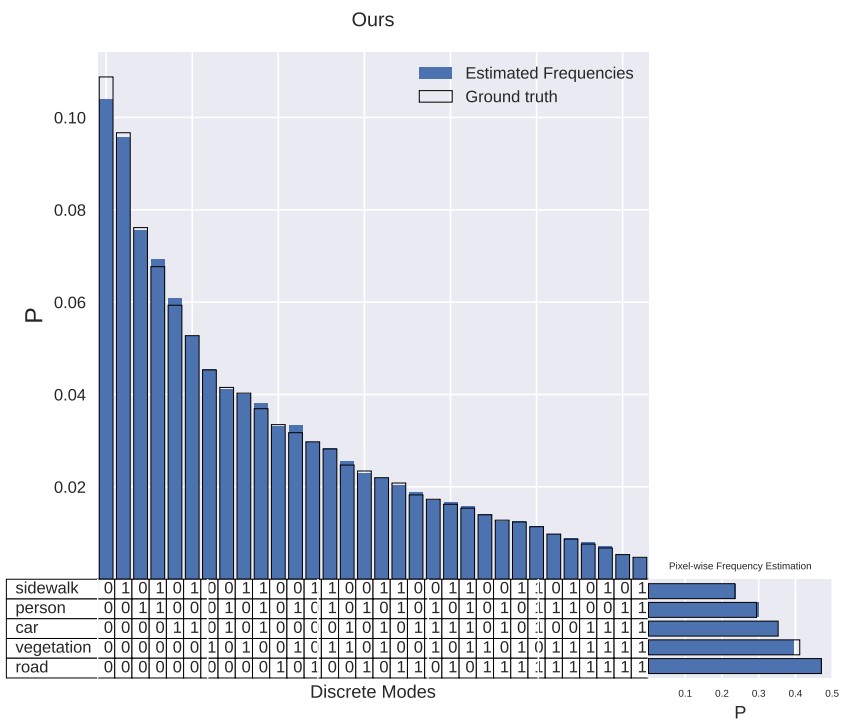

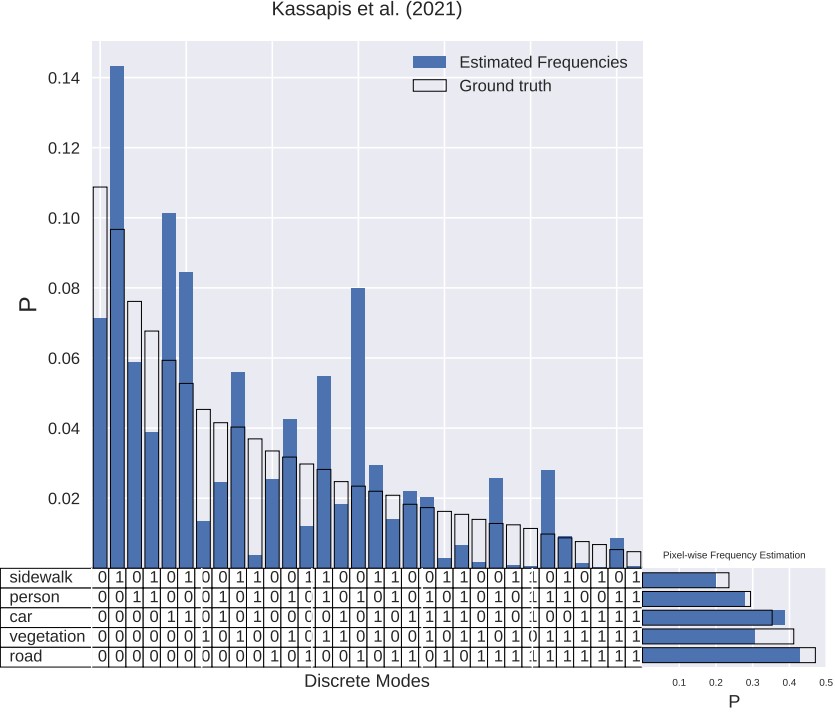

Figure 7: We compare the calibration of our method and Kassapis et al. (2021) in terms of mode-wise probabilities and pixel-wise class flip ratios. In the table below, we use '0' to demonstrate the class is not flipped, and use '1' to demonstrate the class is flipped.

calculate the mode ratio and pixel flip ratio of model outputs. To calculate the mode ratio, we match each output sample to the nearest ground truth label to determine its mode, where the distance is computed as 1-IoU. For the class flip ratio, we count the relative proportions between the five flipped classes and the corresponding original classes for all pixels across all samples. In Figure 7, we show the mode ratio and pixel-wise class flip ratio of our method, and compare with Kassapis et al. (2021). Specifically, we use the weighted output representation with 35 samples (last row in Table 3) and generate the output of Kassapis et al. (2021) by running the pre-trained model provided, and evaluated with 35 samples (row #4 in Table 3). We see that our method has better calibration with the ground truth probabilities at both mode-level and pixel-level.

### D.3 ANALYSIS ON THE TYPE OF UNCERTAINTY

We conduct experiments to analyze the type of predictive uncertainty on the Cityscapes dataset similar to the experiments conducted on the LIDC dataset. We refer to Sec. C.3 for the detail explanation on experimental settings and do not repeat here. Results are evaluated in 35 samples with standard-sampling representation. As shown in Table 11, we get similar observations as in the LIDC where both the uncertainty scores and the evaluation metrics do not change much with the varies in training data size. Therefore, the epistemic uncertainty has less account in the predictive uncertainty of our model.

Table 11: We conduct experiments on the Cityscapes dataset with different training data sizes. Results are evaluated in 35 samples with standard-sampling representation.

| Train dataset | Entropy | Sample Diversity | GED ↓ | M-IOU ↑ | ECE ↓ (%) |
|---|---|---|---|---|---|
| Cityscapes / 4 | $0.823 \pm 0.006$ | $0.598 \pm 0.002$ | $0.157 \pm 0.002$ | $0.686 \pm 0.004$ | $1.878 \pm 0.009$ |
| Cityscapes / 2 | $0.814 \pm 0.004$ | $0.597 \pm 0.002$ | $0.156 \pm 0.001$ | $0.687 \pm 0.005$ | $1.858 \pm 0.013$ |
| Cityscapes | $0.817 \pm 0.010$ | $0.595 \pm 0.004$ | $0.155 \pm 0.002$ | $0.690 \pm 0.002$ | $1.861 \pm 0.015$ |

### D.4 ANALYSIS ON THE NUMBER OF EXPERTS

For the main experiments on the Cityscapes dataset, we assume the ground truth mode number (32) is known and hence choose a slightly larger expert number K = 35. This allows us to validate how the model deals with the redundant experts. If we do not have any prior information on the mode number, we can also use a validation set to draw the curve on how performance changes with the increase of expert numbers and select the K number at the inflection point of the curve.

Below, we show the experimental analysis of the expert number K for model training on the Cityscapes dataset. Specifically, we evaluate the model performance with expert-number K = 25, 30, 32, 35, 40, 50, 80 where 35 is our default expert number. As shown in Figure 8, we see that when the expert number is smaller than 40, with the increasing number of experts, the model achieves a better GED score in general. However, when the expert number is much larger (eg. 50, 80 experts), the model performance decreases, corresponding to an increase of GED score. This phenomenon is similar to experiments on the LIDC dataset where we have discussed in C.4 and do not repeat here. Besides, K = 32 is the inflection point and can be chosen when no prior information is known. Results are evaluated with 35 samples in standard-sampling representation.

### D.5 COMPLEMENTARY QUALITATIVE ANALYSIS

In Figure 9, we show the predictions from all 35 experts. We matched 32 of them with 32 ground truth modes for easy comparison. We see that the expert networks learn the variety of modes well and the gating network gives well-calibrated mode probabilities. Also, we notice that our model learns relatively compact experts, with the extra experts shown at the bottom of the predictions having relatively small weights.

## E DISCUSSION

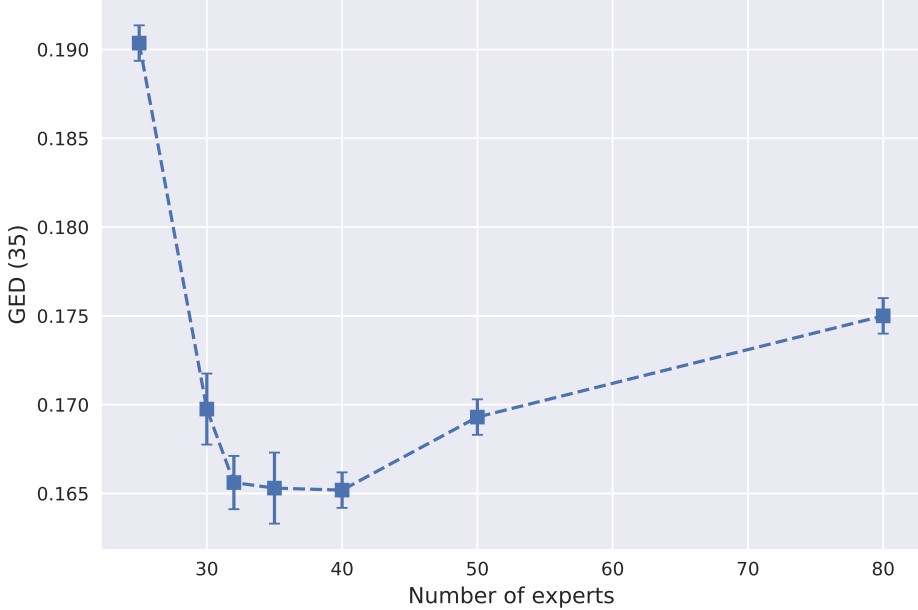

Figure 8: The GED score changes with the number of experts, evaluated with 35 samples in standard-sampling representation. Experiments are conducted on the Cityscapes dataset.

### E.1 WHY IT WORKS BETTER:

We consider the following two factors as the main reasons for the superior performance of our method:

**A flexible multi-modal representation:** We explore an explicit multi-modal uncertainty representation in a form of mixture of stochastic experts, which differs from the previous work with more restricted structures (Kohl et al., 2018; 2019; Baumgartner et al., 2019; Monteiro et al., 2020). Our representation provides a divide-and-conquer strategy for capturing complex multi-modal distributions: each expert learns a clustering of segmentation outputs, which is simpler to model, while the gating network focuses on the overall frequency of those learned modes. This is validated in our empirical results, where our method captures the probability of each mode in Cityscapes much better than previous SOTA (Kassapis et al., 2021).

**Wasserstein-like loss:** We directly minimize the Wasserstein distance between segmentation mask distributions, while previous work either adopt the KL divergence (Kohl et al., 2018; 2019; Baumgartner et al., 2019; Monteiro et al., 2020) or JS divergence (Kassapis et al., 2021) as the loss. Wasserstain distance is known to have better characteristics in measuring the geometry of distributions (Feydy et al., 2019) and mitigate the mode collapse problem (Arjovsky et al., 2017) and show promising effect in the closely-related generative modeling tasks (Arjovsky et al., 2017). We have also conducted additional experiments by replacing the current loss and training pipelines to a VAE style, and demonstrates the superiority of our Wasserstein-like loss (see Appendix C.7).

### E.2 LIMITATIONS:

We discuss some of the limitations of our method in the following: Firstly, when the expert number is larger than the real mode number in the dataset, it is possible to have two expert networks giving similar predictions. As discussed in Sec. A.1, this makes the matching process unstable and we adopt the soft gradient trick to handle this scenario. In the future, efforts can be done by either stabilizing

the matching process or automatically pruning/merging those duplicated experts. Secondly, the matching is based on the cost matrix, however, some existed pairwise loss functions may not provide a good measure for comparison. For example, on the LIDC dataset where the class imbalance scenario exists, using the cross-entropy loss makes the measure dominated by the background pixels, which leads to less satisfactory matching results and further disturbs the training.

### E.3   COMPARE TO ENSEMBLE MODELS

While our MoSE architecture shares certain similarity with the ensemble-based models (i.e., a combination of simpler models), there are three key differences between our framework and the ensemble methods:

**Different goals:**   Most mixture-based ensemble methods aim to learn a predictor with a reduced model variance by aggregating multiple variants of the base models. By contrast, we use the mixture architecture to explicitly capture the multi-modal distribution caused by the aleatoric uncertainty, in which each component corresponds to a different mode. We intend to learn those modes from data without explicit supervision on the mode identity.

**Learning strategy:**   The ensemble methods typically learn each component model based on the original task loss (mostly with a unique groundtruth for each input), and it is usually unclear which part of the data distribution it captures. In contrast, we design a Wasserstein-like loss to minimize the distribution distance between the model outputs and the groundtruth. Thanks to the OT-based formulation, each expert of our model learns to capture a specific mode in the dataset.

**Representation:**   We develop a MoSE structure specifically for our uncertainty modeling task. Our stochastic expert models share most of their parameters and only vary in the latent distributions, which facilitates the learning of compact and meaningful modes. This differs from the conventional design of the component models with totally different parameters in the ensemble methods. Empirically, we have conducted experiments with our OT-based loss replaced by the IoU-loss (cf. Sec. 4.3.1), which is largely equivalent to learning a standard ensemble model. As shown in Table 2 Row #1, this leads to a performance drop of more than 0.3 in GED, demonstrating that a naive ensembling technique may not suffice for this task. We note that Kohl et al. (2018) have also tested the performance of a Unet ensemble method and reached a similar conclusion.

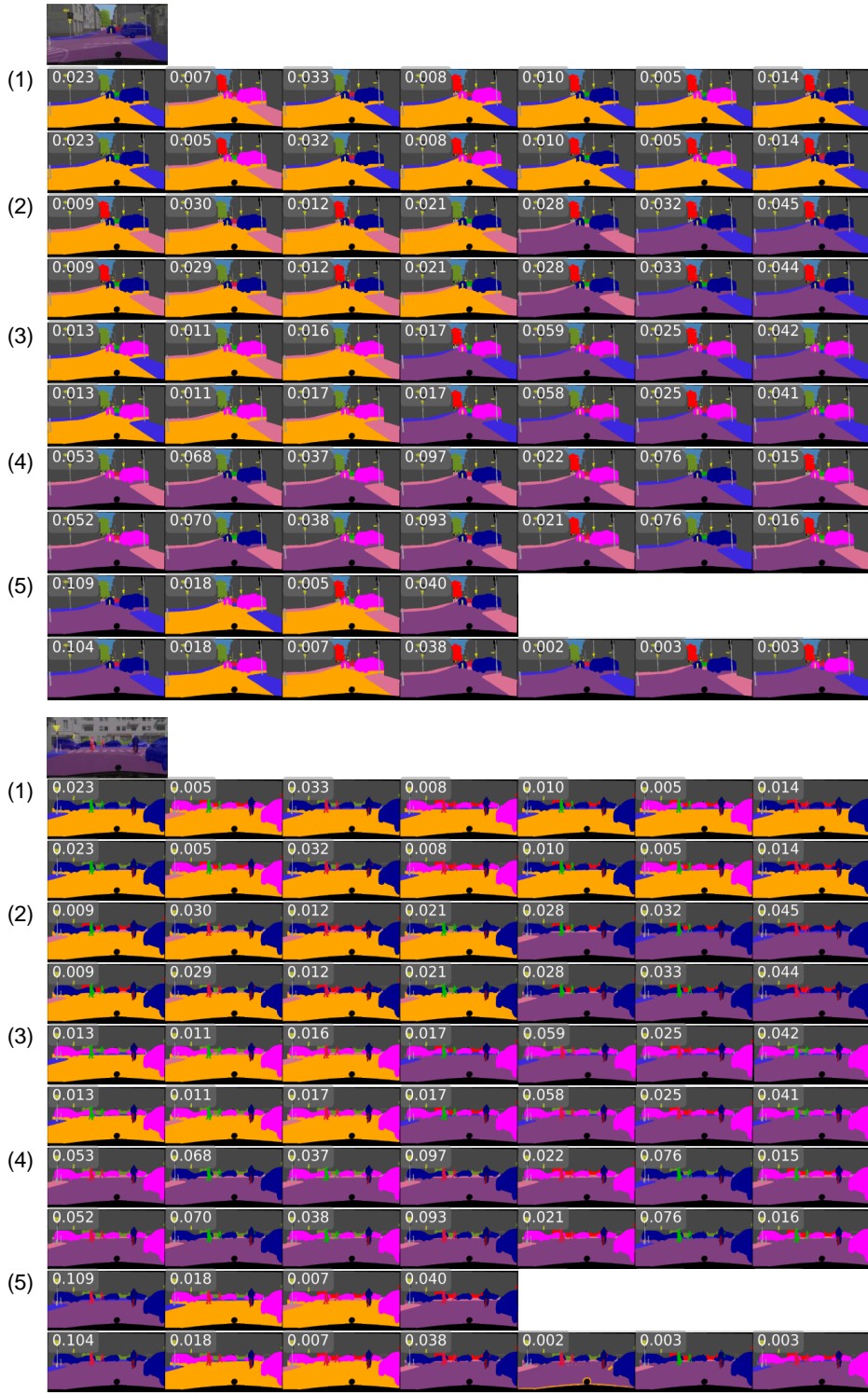

Figure 9: The extra visualization results on the Cityscapes. On the first line of each case, we show the input image masked with a ground truth label. Rest lines show samples from 35 experts and 32 ground truth modes. For clarity, we show matched pairs for ground truth masks on the top and predicted masks on the bottom and divide them into 5 subgroups labeled as (1) - (5). The predictions are generated by sampling once from each expert, we denote the corresponding predicted probabilities on the left head of each prediction.

