# OpenReview forum: "Modeling Multimodal Aleatoric Uncertainty in Segmentation with Mixture of Stochastic Experts"
_ICLR.cc/2023/Conference — ICLR 2023 poster_

### Official Review · Reviewer_bd5h · 2022-10-26

**Confidence:** 4
**Correctness:** 3
**Technical Novelty And Significance:** 2
**Empirical Novelty And Significance:** 2
**Recommendation:** 6

**Clarity, Quality, Novelty And Reproducibility:**

The paper is largely clearly written. There are some missing details or justification of choices which I have pointed out in the main review. While the method in itself is original, the approach and the problem being addressed has issues which cannot be overlooked. Also, there are connections to other existing work which are not explored/discussed.

**Strength And Weaknesses:**

**Strengths:**

* The primary motivation of capturing segmentation uncertainty as a measure of aleatoric uncertainty has been studied widely. This work presents a two-step approach to capture mode level differences (experts) and then finer, pixel level uncertainties per mode. The analogy of treating modes as different experts is reasonable but has issues which are desrcibed further below.

* The experiments are performed on two commonly reported multi-rater datasets, with relevant baselines. The reported results show improved performance for the proposed method.

**Weaknesses:**
* **Expert specific modes**: The assumption that each annotator can be modelled as a mode is not well motivated/justified. This implies that the uncertainty in segmentation is only between raters. In medical image segmentation tasks this is seldom the case. Variations in the data that cause ambiguity in segmentations could be due to other factors such as differences in acquistion, pathologies and intra-rater variability. This work explicitly models and trains each mode to correspond to a different rater, and in doing so it is highly restricted. How would this mixture of experts approach extend to differences arising due to medical data acquired from different centers, for example?

* **Input agnostic, expert specific latent prior**: The latent distributions for $z_k \sim \mathcal{N}(m_k,\sigma_k I)$ is said to be input agnostic with capabilities to model different experts. Specifically, how are the parameters of the latent priors obtained? In Sec 3.2.1 the expert specific prior parameters $m_k$ is mentioned but not how these values are trained. Also, the parameters of the categorical prior over the mixture components is unclear.

* **VAE with Gaussian mixture prior**: The presented model with its categorical prior over the mixture components and then mode specific latents resembles a VAE with Gaussian mixture prior, which has been studied extensively such as in [4]. Is there a specific reason the authors deviate from this established notion of modelling Gaussian mixture type latents within generative model settings?  To me this model reduces to the probabilistic U-net with a Gaussian mixture prior [5]; making this connection to [4] and [5] also could help with contextualising the contribution.

* **Gating module**: The key point of departure from [4,5] is the gating network; the role of which is unclear to me. Is the gating module injecting the input image features to choose which expert should be chosen? If yes, is the rater specific probabilities based on the input image?

* **What happens when K=1**: In instances when there are no multiple raters to train from i.e., when K=1, does this model still capture aleatoric uncertainty? For instance, in [2,5] experiments when trained with only a single rater also show the model is capable of optimising for metrics such as GED.

* **GED as the main metric**: GED is a distribution matching score which is trained to match the diversity in annotation variability and the predicted segmentations. This work primarily focuses on matching the annotator diversity as a means to quantify the uncertainty. While this is also what most existing literature is doing, there are concerns on the usefulness of this approach. For instance, this work in [1] takes up the usefulness of segmentation measures and clearly shows that GED is not a very useful measure. Using sample diversity to be somehow a measure of meaningful uncertainty is problematic, in my opinion. And I think as a community we need to move away from this notion and think about more useful ways of quantifying segmentation uncertainty, based on some of the discussions in [1].


[1] Mehta, Raghav, Angelos Filos, Ujjwal Baid, Chiharu Sako, Richard McKinley, Michael Rebsamen, Katrin Dätwyler et al. "QU-BraTS: MICCAI BraTS 2020 Challenge on Quantifying Uncertainty in Brain Tumor Segmentation--Analysis of Ranking Metrics and Benchmarking Results." arXiv preprint arXiv:2112.10074 (2021).

[2] Christian F Baumgartner, Kerem C Tezcan, Krishna Chaitanya, Andreas M Hötker, Urs J Muehlematter, Khoschy Schawkat, Anton S Becker, Olivio Donati, and Ender Konukoglu. Phiseg: Capturing uncertainty in medical image segmentation. In International Conference on Medical Image Computing and Computer-Assisted Intervention, 2019.

[3] Bø, Hans Kristian, et al. "Intra-rater variability in low-grade glioma segmentation." Journal of neuro-oncology 131.2 (2017): 393-402.

[4] Dilokthanakul, Nat, et al. "Deep unsupervised clustering with gaussian mixture variational autoencoders." arXiv preprint arXiv:1611.02648 (2016).

[5] Kohl, Simon, Bernardino Romera-Paredes, Clemens Meyer, Jeffrey De Fauw, Joseph R. Ledsam, Klaus Maier-Hein, S. M. Eslami, Danilo Jimenez Rezende, and Olaf Ronneberger. "A probabilistic u-net for segmentation of ambiguous images." Advances in neural information processing systems 31 (2018).


**Summary Of The Paper:**

This work presents a two-step process to model the aleatoric (data) uncertainty in image segmentation using a mixture of experts type of an approach. The main assumption in this work is that each expert could correspond to a mode in the true, underlying uncertainty distributions that are multi-modal. Assuming the number of modes/experts to be known, a stochastic neural network predicts segmentation masks. A second gating network trained to mimic the expert behaviour by predicting the probabilities of experts. Experiments on two common multi-annotator datasets are reported and compared to relevant baseline methods using measures such as generalized energy distance (GED).

**Summary Of The Review:**

**Edit (Scores Update)**: Updating from 3 to 6.

Capturing segmentation uncertainty that is well calibrated can be useful in many applications, such as in medical image analysis. This work presents a two-step approach of modelling expert level behaviour and then within a mode capturing finer pixel-level uncertainties. The use of existing metrics such as GED is problematic. Also the connections to VAEs with Gaussian mixture priors is entirely missing.

---

> ### Author Response · Authors · 2022-11-19
> **Response to Reviewer bd5h - Part 3**
>
> **Response to weakness 5 on K=1:**
>
> We would like to point out that the notation K in our paper denotes the number of expert used in our MoSE model (cf. Sec 3.2), and is different from the number of raters (which is potentially an unknown value),  as well as the number of annotations per image M (cf. Sec 3.1).
>
> We have conducted the experiments with the number of annotations per image M set to 1 on the LIDC datast (cf. Sec 4.1), which is the same experiments that previous work [1,2] conducted. We note that this setting does not mean the rater number is one since different images could still be annotated by different raters. Since the underlying distribution is still multi-modal, we keep use K=4 experts for the one-annotated experiments. The comparison results are shown in Table 1. Our GED score (0.223) is better than Kohl et al.(0.445)  and Baumgartner et al. (0.323). This demonstrates the superiority of our method in dealing with limited annotations.
>
> We note that the case with a unique rater was not studied in the experiments of related work [1,2]. Under such a setting, our method can still capture intra-rater variation, whether it is multi-modal or not. In fact, as our loss optimizes the distance between the predicted distribution and the ground truth distribution (See Eq. 5), our model is also capable of modeling the aleatoric uncertainty for the single rater case.
>
> [1] Christian F Baumgartner et al. Phiseg: Capturing uncertainty in medical image segmentation. MICCAI, 2019.
> [2] Kohl S et al. A probabilistic u-net for segmentation of ambiguous images. NIPS, 2018.
>
> **Response to weakness 6 on the metric:**
>
> We respectively disagree with the reviewer on the metric used in our work. We emphasize that we focus on predicting a segmentation distribution as a means to represent the uncertainty and achieve this by matching our predicted distribution to the (sampled) groundtruth annotation distribution. To evaluate how well the distribution is captured, we adopt three different metrics: GED, M-IoU and ECE, which measure the uncertainty quantification from various aspects and serve as our metric system as a whole.
>
> Specifically, the generalized energy distance (GED) measures the statistical distance between two variables, which equals zero if and only if two variables are identically distributed [2]. As shown in Eq. (8) of Appendix C, if the two variables have different distributions but with the same diversity, the GED score could still be large.  Besides, the energy score has been widely adopted as a distributional measure loss in generative modeling literature[3,4], demonstrating its efficacy in measuring the distributional distance. One limitation of the  GED  metric is that it may have a bias towards sample diversity in some cases, and we have explicitly pointed out this in our paper (cf. Sec. 4.4, Para. 2, line:3-4 and Appendix A.2). To remedy this, we use the other two metrics M-IoU and ECE score which are not biased by diversity.  The Matched-IoU score calculates an average IoU score, which measures a distributional distance and is not biased by sample diversity [5]. Besides, we also adopt the Expected Calibration Error (ECE) metric, which is widely adopted in the general uncertainty calibration problem [6] and has also been used in the segmentation task [7]. It measures the uncertainty in a pixel-wise manner and is as well unbiased for sample diversity.
>
> We thank the reviewer for pointing out the work on developing an evaluation score on uncertainty quantification (QU-BraTS). However, we note that the metric proposed by [1] mainly focuses on the uncertainty measure with a single prediction. Hence it is not capable of measuring the spatially correlated and potentially multi-modal uncertianty in our problem, which is also discussed by [1] (see the last 3rd paragraph in Sec. 6). We agree that exploring more useful evaluation metrics for the segmentation uncertainty is critical and will leave this to the future work.
>
> [1] Mehta Raghav et al. QU-BraTS: MICCAI BraTS 2020 Challenge on Quantifying Uncertainty in Brain Tumor Segmentation--Analysis of Ranking Metrics and Benchmarking Results. arXiv preprint, 2021.
> [2] Székely G J, Rizzo M L. Energy statistics: A class of statistics based on distances. Journal of statistical planning and inference, 2013.
> [3] Bellemare M G et al. The cramer distance as a solution to biased wasserstein gradients. arXiv preprint, 2017.
> [4] Arbel M et al. On gradient regularizers for MMD GANs. NIPS, 2018.
> [5] Simon AA Kohl et al. A hierarchical probabilistic u-net for modeling multi-scale ambiguities. NIPS Workshop, 2019.
> [6] Chuan Guo et al. On calibration of modern neural 384 networks. ICML, 2017.
> [7] Jungo A, Reyes M et al. Assessing reliability and challenges of uncertainty estimations for medical image segmentation. MICCAI, 2019.

---

> > ### Comment · Reviewer_bd5h · 2022-12-12
> > **Authors clarify most concerns**
> >
> > I thank the authors for their thorough responses to my first set of reviews. I stand corrected on some of the concerns raised which have now been clarified.
> >
> > However, I continue to maintain the broader concern about uncertainty quantification using metrics such as GED. Further, the first version of the submission was a bit unclear (perhaps, resulting in some of the ambiguities).
> >
> > Taking the responses to other reviews and to my own, I will raise my score to 6 (weak accept) and commend the authors on their effort.

---

> > > ### Author Response · Authors · 2022-12-12
> > > **Thanks for your response!**
> > >
> > > Thank you for raising your score to a weak accept and for your continued engagement with our paper. We appreciate your feedback and are glad that our responses were able to clarify some of the concerns you had.
> > >
> > > Regarding your concern about uncertainty quantification using metrics such as GED, we agree that GED is not a perfect measure of uncertainty. However, we believe that it is still a useful metric for evaluating the uncertainty of our model, especially in comparison to other models. We have carefully considered the limitations of GED and have discussed them in the paper. As a result, we have also investigated two other metrics, M-IoU and ECE, to evaluate the quality of our uncertainty quantification. In future work, we plan to further explore alternative metrics of uncertainty quantification.
> > >
> > > Thank you again for your constructive feedback and for your recognition of our efforts. We will continue to work on improving the clarity and rigor of our paper.

---

> ### Author Response · Authors · 2022-11-19
> **Response to Reviewer bd5h - Part 2**
>
> **Response to weakness 3 on VAE with GMM prior:**
>
> We thank the reviewer for mentioning the related work in VAE with gaussian mixture prior. We discuss the difference below and add relevant content in the revised paper.
> - Compared to the VAE model family, our major difference is the use of implicit density models, where we capture each mode in the label distribution by mapping from an expert-specific gaussian noise. This enables us to adopt the Wasserstein distance as the loss function, which has better characteristics in measuring the geometry of distributions [2] and mitigates the mode collapse problem [1], and shows promising effects in the closely-related generative modeling tasks [1].
> - In addition, our model does not need to train an additional posterior network as in VAE-based methods, which enables us to build a lightweight model with negligible parameters added to the original segmentation backbone (See Table 1 or Appendix Table 4). Notably, the total parameter number in our model is only 55% of the VAE-based method [3,4] (See Appendix Table 4 or Table B1 below).
> - Finally, we develop a specific design for our mixture model -- instead of conditioning all the latent variables on the input as in Probabilistic U-net [3], we adopt a factorized design where the prior gaussian distribution is input agnostic and the categorical distribution is input dependent. This regularizes the learned distribution and encourages the model to learn some shared patterns in the dataset,  thus producing a meaningful two-level uncertainty representation.
>
> We also make an empirical comparison between our method and VAEs with GMM prior [5,6,7]. To do this, we replace the Gaussian prior distribution in Probabilistic U-net [3] by a GMM with K components and approximate the ELBO via Monte Carlo sampling as in [5]. Besides, we also evaluate a model variant by using input-agnostic Gaussian distributions and an input-dependent categorical distribution as our model. We conduct experiments on the LIDC dataset, use K = 4  Gaussian distributions (same as our expert number), and keep other settings the same as the Probabilistic U-net [3]. The results are summarized in Table B1. We observe that the GMM prior does improve the original Probabilistic U-net (ProbUnet) [3]. However, our model still achieves the best performance, outperforming other models with a large margin (eg. 3% in GED). This demonstrates the efficacy of our design, especially our OT-based loss (for being one of the major differences).
>
> Table B1. An empirical comparison of our method and VAEs with GMM prior. We use the reimplemented version of ProbUnet provided by Baumgartner et al. [4] to keep the same Unet backbone.
>
> | Method                      | Gaussian Prior(s) | GED ↓        | M-IOU ↑      | ECE（%）↓    | #param. |
> |-----------------------------|-------------------|--------------|--------------|--------------|---------|
> | ProbUnet                    | Input dependent   | 0.298 ±0.010 | 0.527 ±0.007 | 0.118 ±0.012 | 76.15   |
> | ProbUnet with GMM prior (1) | Input dependent   | 0.254 ±0.008 | 0.597 ±0.005 | 0.111 ±0.013 | 76.18   |
> | ProbUnet with GMM prior (2) | Input agnostic    | 0.251 ±0.004 | 0.597 ±0.007 | 0.101 ±0.010 | 76.14   |
> | Ours                        | Input agnostic    | **0.218 ±0.003** | **0.624 ±0.004** | **0.064 ±0.015** | 41.60   |
>
> [1] Arjovsky M et al. Wasserstein generative adversarial networks. ICML, 2017.
> [2] Feydy J et al. Interpolating between optimal transport and mmd using sinkhorn divergences. AISTATS, 2019.
> [3] Kohl S et al. A probabilistic u-net for segmentation of ambiguous images. NIPS, 2018.
> [4] Baumgartner C F et al. Phiseg: Capturing uncertainty in medical image segmentation. MICCAI, 2019.
> [5] Lee D B et al. Meta-gmvae: Mixture of gaussian vae for unsupervised meta-learning. ICLR, 2021.
> [6] Dilokthanakul N et al. Deep unsupervised clustering with gaussian mixture variational autoencoders. arXiv preprint, 2016.
> [7] Zhuxi Jiang et al. Variational deep embedding: An unsupervised and generative approach to clustering. IJCAI 2017.
>
> **Response to weakness 4 on the gating module:**
>
> Yes, the gating network takes the image as input and predicts the probabilities on which expert should be chosen. However, it does not represent the rater probabilities. As we discussed in the response to weakness 1, our expert network (or mode) does not correspond to the raters. For example, as shown in Figure 2, the rater probabilities are always [1/4,1/4,1/4,1/4]; however, the learned expert probabilities vary for different input images. The output probabilities of the gating network, also denoted in categorical distribution $\mathcal{G}(\boldsymbol\pi)$, represent an important characteristic of the multi-modal distribution. We have also validated its efficacy in our ablation study (cf. Table 2. row #2).

---

> ### Author Response · Authors · 2022-11-19
> **Response to Reviewer bd5h - Part 1**
>
> We thank the reviewer for the time and comments. However, there are several unfortunate misunderstandings about our method, which we want to clarify in the following.
>
> Summary: We want to point out that our general framework is not a two-step approach, and all the model parameters are end-to-end learned via the proposed OT-based loss. Also, we only assume the underlying uncertainty distribution to be multi-modal. We do not assume the number of modes to be known, neither do we assume that each learned mode corresponds to a specific annotator.
>
> **Response to weakness 1 on expert-specific modes:**
>
> We respectfully disagree with the reviewer on the expert-specific mode assumption. Our method does not assume each mode corresponds to a rater. Neither do we train each expert to model a distinctive rater. During training, we do not have information on the raters of annotations, and our MoSE model learns to cluster the segmentation masks into distinctive modes, which is largely data-driven. For example, **it is possible that different raters with similar annotations are captured by the same expert.** (See Figure 2 (right) as an example, the 1st, 3rd and 4th annotations given by different raters correspond to mode #0 in the predicted samples.) In addition, **annotations from a rater on different images could also be matched into different modes.** It is common that a rater gives a blank annotation for one image and a non-blank annotation for another, which could correspond to different modes in our model.
>
> While the aleatoric uncertainty in segmentation could be caused by a variety of factors including the intra-rater and inter-rater variability, we believe our method is capable of capturing those variations from different sources. Empirically, the LIDC dataset we used is a multi-center dataset containing four annotations per image from a total of twelve anonymized radiologists (See Sec 4.1 and [1]). In this setting, the variation in annotations could contain different sources, and we validate that our method achieves SOTA on capturing the aleatoric uncertainty (See Table 1) and learning meaningful modes (See Figure 2).
>
> [1] Samuel G Armato III et al. The lung image database consortium (lidc) and image database resource initiative (idri): a completed reference database of lung nodules on ct scans. Medical physics, 2011.
>
> **Response to weakness 2 on latent priors:**
>
> The parameters for latent priors are randomly initialized and end-to-end trained via our Optimal Transport-Based loss as explained in Sec. 3.3. Specifically, as shown in Eq. (6),  we solve an Optimal Transport problem to match among samples and annotations, and perform backpropagation to jointly optimize $\theta_s$ and $\theta_{\pi}$. We note that, as defined in Sec. 3.2.1,   the parameters of latent priors ${m_k, \sigma_k}$ with $k=1\cdots K$ are included in $\theta_{s}$, and the commonly-used reparameterization trick for Gaussian distribution is adopted to enable backpropagation [1].
>
> The parameters of the categorical distribution $\boldsymbol{\pi}$ are predicted via the gating network as explained in Sec. 3.2.1. It is also trained using our OT-based loss (see Sec. 3.3), where all the related parameters are also included in $\theta_{\pi}$.
>
> [1] Diederik P Kingma and Max Welling. Auto-encoding variational Bayes. ICLR, 2014

---

> ### Author Response · Authors · 2022-12-05
> **Gentle Reminder**
>
> Dear Reviewer bd5h,
>
> Thanks again for your time and valuable feedback!
>
> As the deadline for the discussion period is approaching, we would be grateful if you could check our responses with the updated manuscript, and confirm whether our responses have addressed your concerns.
>
> Please let us know if there are any further questions or suggestions.  We will do our best to respond to them.
>
> Best regards,
> Authors of Paper 1385

---

> ### Author Response · Authors · 2022-12-12
> **Your feedback is important to us**
>
> Dear Reviewer bd5h,
>
> Thank you so much for taking the time to review our paper. We truly appreciate your constructive comments and are grateful for the effort you put into reviewing our work.
>
> In our initial response, we addressed some misunderstandings you had about our method and the evaluation metrics we adopted. We also clarified our connections to VAEs with Gaussian mixture priors and revised our related work section to reflect this.
>
> We would be grateful if you could let us know if our response addresses your concerns.  Your feedback is really valuable to us, and we will treat it earnestly.
>
> Best regards,
> Authors of Paper 1385

---

### Official Review · Reviewer_LSxs · 2022-10-26

**Confidence:** 3
**Correctness:** 3
**Technical Novelty And Significance:** 3
**Empirical Novelty And Significance:** 3
**Recommendation:** 6

**Clarity, Quality, Novelty And Reproducibility:**

The paper is well-written and easy to follow. Most details for reproducibility are included in the paper.

**Strength And Weaknesses:**

Strengths:
- The paper is well written and easy to follow. It tackles an important problem of capturing uncertainty in segmentations.
- The paper proposes a simple yet novel method of combining multi-head predictions (which reminds me of [1]) together with a gating network to capture the distribution of segmentation predictions.
- The paper compares to relevant baselines on standard benchmarks and shows competitive results.

Weaknesses:
- It would be interesting to compare the proposed method to an ensemble of predictive methods that would be able to capture multi-modality.

Misc:
- Why are methods like the Prob. U-Net incapable of capturing multi-modality?
- What's the overhead of this method compared to a single prediction head? Is the comparison fair given that the method might have more parameters etc?
- Do you have an intuition why "compact" performs better than the "standard sampling"?

[1] Osband, Ian, et al. "Deep exploration via bootstrapped DQN." Advances in neural information processing systems 29 (2016).

**Summary Of The Paper:**

The paper proposes a novel method for modelling predictive uncertainties for segmentation tasks. The model consists of an encoder-decoder architecture with multiple decoder heads that are weighted by a separate gating network that predicts the weighting based on the encoded image code. The method is compared on the LIDC and cityscapes benchmarks and exhibits competitive performance.

**Summary Of The Review:**

The paper proposes a simple yet novel and effective method at capturing the uncertainty segmentation predictions. The experiments support the claims of the paper.

---

> ### Author Response · Authors · 2022-11-19
> **Response to Reviewer LSxs**
>
> **Response to the weakness:**
>
> We discuss the main differences between our method and ensemble methods in the general response above. Typically, while an ensemble of predictive methods may be able to capture multi-modality, it is difficult to learn a compact representation where each predictive model corresponds to a single mode.
>
> Some other ensembling-based methods explicitly divide the training dataset according to external knowledge (eg. the expert identification in [1]) and learn a set of mode-specific experts. However, they require full supervision for the modes, which is unavailable in our problem (cf. Sec 4.1).
>
> **Response to misc 1 on Prob. U-net and multi-modality:**
>
> Empirically, we found that previous methods like the Prob. U-Net have a limited capacity in capturing multi-modality, in the sense that the proportions of distinctive modes often misalign with the groundtruth. This is potentially caused by the over-regularization from the diagonal Gaussian prior distribution[2], or other approximation problems in VAEs, like the posterior collapse [3]. Therefore, our method explicitly represents the multi-modal characteristic via the mixture of experts framework. Such a strategy leads to a better estimation on the mode probabilities in our experiments.
>
> **Response to misc 2 on overhead of our model:**
>
> Compared to a single-head segmentation network, our model primarily adds a three-layer MLP for the gating network and K diagonal Gaussian prior models for the K stochastic experts. Typically, these additional modules are lightweight and in our experiments, the number of their parameters equals 0.8% of the parameters in the Unet backbone (cf. Appendix Tabel 4). Moreover, each new expert component (or head) only adds a negligible number of parameters including a new gaussian prior model parameterized by $m_k, \sigma_k\in R^L$, and an additional output channel for the gating network (cf. Sec 3.2.1).
>
> As demonstrated in the general response, the amount of parameters in our model is smaller or comparable to the baseline models, which ensures fairness in our comparison.
>
> **Response to misc 3 on compact and standard sampling:**
>
> The difference between "compact" and "standard sampling" is that the former uses a set of weighted samples to represent the model output where the weights correspond to the mode probabilities, while the latter performs a standard sampling to generate a set of outputs, which has an additional sampling step from the mode probabilities (i.e. the gating weights). When the sample size N is relatively small compared to the number of modes/experts, the additional sampling step in "standard sampling" could generate a noticeable amount of variations [4], leading to inaccurate representations of the mode distribution and worse performance. We also note that with an increasing number of samples, the performance gap decreases. As shown in Table 1, the performance gap is 2.3% for 16 samples and decreases to 0.3% for 100 samples.
>
> We emphasize that our compact form is novel and provides an efficient way to represent the multi-modal segmentation uncertainty, and is especially useful for realistic scenarios where the number of modes is large.
>
> [1] Guan M et al. Who said what: Modeling individual labelers improves classification. AAAI, 2018.
> [2] Dilokthanakul N et al. Deep unsupervised clustering with gaussian mixture variational autoencoders. arXiv preprint, 2016.
> [3] Ali Razavi et al. Preventing posterior collapse with delta-vaes. In ICLR, 2018.
> [4] Thompson S K. Sample size for estimating multinomial proportions. The American Statistician, 1987.

---

> > ### Comment · Reviewer_LSxs · 2022-12-12
> > **Thanks for clarifications**
> >
> > I thank the authors for the thorough rebuttal and response to my questions as well as to those of the other reviewers. So far all my questions have been answered and I'll engage in discussions with the other reviewers.

---

> > > ### Author Response · Authors · 2022-12-13
> > > **Thanks for your response!**
> > >
> > > Thank you for your positive response to our rebuttal and for engaging in further discussions with the other reviewers. We are delighted that we were able to address your questions and concerns. We look forward to receiving any further feedback you may have and to continuing to collaborate with you to enhance our manuscript.

---

> ### Author Response · Authors · 2022-12-05
> **Gentle Reminder**
>
> Dear Reviewer LSxs,
>
> Thanks again for your time and valuable feedback!
>
> As the deadline for the discussion period is approaching, we would be grateful if you could check our responses with the updated manuscript, and confirm whether our responses have addressed your concerns.
>
> Please let us know if there are any further questions or suggestions. We will do our best to respond to them.
>
> Best regards,
> Authors of Paper 1385

---

> ### Author Response · Authors · 2022-12-12
> **Your feedback is important to us**
>
> Dear Reviewer LSxs,
>
> Thank you so much for taking the time to review our paper. We truly appreciate your constructive comments and are grateful for the effort you put into reviewing our work.
>
> According to your suggestion, we have compared our method with the ensembling-based models and added relevant discussion in the revised paper. We have also demonstrated the lightweight of our model and comparability to others by showing the model size of all the benchmark methods in Tables 1 and 3 of the revised paper. For your other concerns, we have also made explanations to them point by point.
>
> As we are hearing the rebuttal is closing up today, we would be grateful if you could let us know if our response addresses your concerns. Your feedback is really valuable to us, and we will treat it earnestly.
>
> Best regards,
> Authors of Paper 1385

---

### Official Review · Reviewer_VqFH · 2022-10-27

**Confidence:** 4
**Correctness:** 3
**Technical Novelty And Significance:** 3
**Empirical Novelty And Significance:** 3
**Recommendation:** 6

**Clarity, Quality, Novelty And Reproducibility:**

*Clarity:*
The paper is well structured and presented in a accessible fashion.

*Quality:*
I critic the missing comments on the comparability with the benchmark models, this needs to be enhanced. The authors conducted many experiments to back their claims and provide extensive supplementary material that clarifies many questions during reading.

*Novelty:*
The proposed method is as far as I know novel in the context of segmentation uncertainty. While Wasserstein losses and the method of constraint relaxation have been used before in multi class segmentation I am not aware of their application in the context of aleatoric segmentation uncertainty.

*Reproducibility:*
The authors publish their code upon publication.

**Details Of Ethics Concerns:**

No ethical concerns.

**Strength And Weaknesses:**

**Strengths:**
- The method is novel in the context of segmentation uncertainty and well presented
- Good structure of the paper
- comparison to many benchmark models
- Results show improvement

**Weaknesses:**
- Not really clear if the benchmark models are comparable in terms of number of parameters etc. Are the benchmark models own implementations?
- Not really clear whether the constraint relaxation leads theoretically to the same result as the original problem formulation. Can you clarify why those choices are justified beyond the fact, that they make the problem calculation tractable?
- There is no discussion of the model beyond two limitations in the appendix. Why does it work?
- No reference to previous use of optimal transport based losses in segmentation, e.g.  *Fidon, Lucas, Sébastien Ourselin, and Tom Vercauteren. "Generalized wasserstein dice score, distributionally robust deep learning, and ranger for brain tumor segmentation: BraTS 2020 challenge." International MICCAI Brainlesion Workshop. Springer, Cham, 2020.*

**Questions:**

- Sec. 3.1. In practice the relative frequency of each annotation is always 1/n in your model since it is quite unlikely that you get the exact same annotation from different experts, so v_n is always a uniform distribution, assigning same probability to all ground truth annotations. From an information theoretic point of view v_n contains no information. Therefore, all information must be inferred from the ground truth segmentation masks over the cost function.
Can you elaborate on how the gating network still learns to predict probabilities that correspond to the relative mode frequencies? What do those probabilities express then?

- How would your model behave in the limit case of many experts K? Do the curves in Fig. 5 and Fig. 8 keep decreasing?

- I would be interested into a row of Table 2 where you test  stochastic experts, uniform expert weights and the IoU-loss.

- Can you elaborate if there is a relation to ensemble models, especially if you set S=1?

- Sec. 4.2 In your experiments for LIDC you set the experts to 4 (equal to the number of annotations available per image) while you set the number of experts to 35 for the Cityscape dataset. This corresponds to the number of ground truth annotations for LIDC and approximately to the number of classes in Cityscapes. How do you justify the assumption that each ground truth annotation (or class in Cityscapes) contains exactly one mode?

- Where do you draw the line between variation around one mode and different mode? For example in LIDC one could argue that one hypothesis is that there is no cancer and the other is that there is cancer and that these should be the modes and that all variation resulting from the experts uncertainty in drawing the segmentation mask should come from the variation around the second mode.

- How do you treat borderline cases of IoU in the cost function matrix? For example empty segmentation mask compared to segmentation mask that contains a mask?

- What is the intuition why the model works better than the benchmarks? One major factor I see is that during training you use all ground truth annotations per image in one gradient step, which is a main difference to the compared models. The paper would benefit from a discussion of the method a lot.

**Summary Of The Paper:**

This paper presents a new architecture for modelling the quantification of aleatoric segmentation uncertainty, i.e. predicting the distribution of segmentations in a given task. The model is tested on two established  datasets. Specifically, the author presents a mixture of stochastic experts model whose parameters are optimised by minimising a loss function based on the optimal transport problem. The coupling is calculated between the relative frequency of the ground truth annotations and the predicted probability for an expert given by the gating network. For the cost of transportation the IoU is used. To enable backpropagation, the original formulation of the optimal transport problem is substituted by an objective derived by constraint relaxation.
The author finds that their model outperforms all benchmark models in terms of common performance measures used in this domain of research on the LIDC dataset, as well as on some for the Cityscape dataset.

**Summary Of The Review:**

The presented method is to the best of my knowledge novel and the authors present it well. However, the paper would benefit from a more detailed discussion and a provided intuition of why the proposed method is superior to the benchmark models. I encourage the authors to address the questions posed above.

---

> ### Author Response · Authors · 2022-11-19
> **Response to Reviewer VqFH - Part 5**
>
> **Response to question 7 on borderline case of IoU cost**
>
> For computing the IoU-based cost matrix, we adopt the common label smoothing technique to cope with the borderline cases. In particular, we use a smoothed IoU loss with '1' added to the numerator and denominator to avoid disappearing gradients for empty segmentation mask cases. Here the loss can be written as $\text{IoU}(p,g) = 1-\frac{\sum_{i=1}^N p_i g_i+1}{\sum_{i=1}^N p_i+\sum_{i=1}^N g_i -\sum_{i=1}^N p_i g_i +1}$, where $p_i$and $g_i$are the $i$-th pixel of the softmax prediction and ground truth label, and $N$denotes the total number of pixels in an image. We note that when the segmentation mask g is empty, the loss becomes $\text{IoU}(p,\mathbf{0}) = 1-\frac{1}{\sum_{i=1}^N p_i+1}$, which encourages the prediction to be all zeros too.
>
> **Response to question 8 on why our model works better**
>
> We consider the following two factors as the main reasons for the superior performance of our method:
>
> 1) A flexible multi-modal representation: We explore an explicit multi-modal uncertainty representation in a form of mixture of stochastic experts, which differs from the previous work with more restricted structures [1,2,3]. Our representation provides a divide-and-conquer strategy for capturing complex multi-modal distributions: each expert learns a clustering of segmentation outputs, which is simpler to model, while the gating network focuses on the overall frequency of those learned modes. This is validated in our empirical results, where our method captures the probability of each mode in cityscapes much better than previous SOTA [4] (cf. Figure 7 in Appendix D.2).
>
> 2) Wasserstein-like loss: We directly minimize the Wasserstein distance between segmentation mask distributions, while previous work either adopt the KL divergence[1,2,3] or JS divergence[4] as the loss.  Wasserstein distance is known to have better characteristics in measuring the geometry of distributions [6] and mitigate the mode collapse problem [5], and show promising effect in the closely-related generative modeling tasks [5]. We have also conducted additional experiments by replacing the current loss and training pipelines to a VAE style, and demonstrates the superiority of our Wasserstein-like loss (see the response to Reviewer 3, weakness 3 or Appendix C.7).
>
> We further analyze the impact of the number of groundtruth annotations used in each training step. To this end, we conduct an ablation study where we random sample one ground truth label in each gradient step on the LIDC dataset and compare the results in Table A3. We observe that the performance of our method remains almost the same for such a setting, which indicates that our advantage does not come from using all groundtruth annotations in each gradient step. Moreover, we have also performed experimental evaluation using one annotation per image for training (see Table 1 with # label = 1) in the original submission. We note that our method still achieves the best performance.
>
> Table A3. Ablation study on using only one ground truth label per gradient step. Experiments are conducted on the LIDC dataset with full annotations and evaluated with 16 samples.
>
> | #gt per gradient step | GED ↓         | M-IoU ↑       | ECE (%) ↓         |
> |-----------------------|--------------|--------------|--------------|
> | One                   | 0.218 ±0.003 | 0.627 ±0.005 | 0.071 ±0.012 |
> | All (four)            | 0.218 ±0.003 | 0.624 ±0.004 | 0.064 ±0.015 |
>
> [1] Simon Kohl et al. A probabilistic u-net for segmentation of ambiguous images. NIPS, 2018.
> [2] Christian F Baumgartner et al. Phiseg: Capturing uncertainty in medical image segmentation. MICCAI, 2019.
> [3] Monteiro et al, Stochastic segmentation networks: Modelling spatially correlated aleatoric uncertainty, NeurIPS 2020.
> [4] Elias Kassapis et al. Calibrated adversarial refinement for stochastic semantic segmentation. ICCV, 2021.
> [5] Arjovsky M et al. Wasserstein generative adversarial networks. ICML, 2017.
> [6] Feydy J et al. Interpolating between optimal transport and mmd using sinkhorn divergences. AISTATS, 2019.
> [7] Fidon Lucas et al. Generalised wasserstein dice score for imbalanced multi-class segmentation using holistic convolutional networks. MICCAI Brainlesion workshop, 2017.
> [8] Fidon, Lucas et al. Generalized wasserstein dice score, distributionally robust deep learning, and ranger for brain tumor segmentation: BraTS 2020 challenge. MICCAI Brainlesion Workshop. 2020.
> [9] Samuel G Armato III et al. The lung image database consortium (lidc) and image database resource initiative (idri): a completed reference database of lung nodules on ct scans. Medical physics, 2011.

---

> > ### Comment · Reviewer_VqFH · 2022-11-28
> > **Response to Authors**
> >
> > I thank the authors for their rebuttal. I found most of my comments addressed. Even though the authors answered the question why their methods works superior to the benchmarks, it stills remains difficult to get an intuition why the method works so well. I believe the paper would be stronger if it would provide a motivation for the design choices. I therefore keep my score.

---

> > > ### Author Response · Authors · 2022-12-01
> > > **Response to Reviewer VqFH**
> > >
> > > Thanks for your reply. We want to point out that we have outlined the motivation for our design choices in the introduction (See para. 2-4). In the following, we provide more details on our motivation and intuitions on why our method works well.
> > >
> > > **Motivation:** We observe that the aleatoric uncertainty in existing datasets typically has a multi-modal structure (eg. the blank and non-blank annotations on the LIDC dataset), where different modes correspond to the major disagreements in image interpretation. For each image, its mode proportions encode the main aspect of uncertainty, which is important for calibration or downstream decision-making in practice. Most previous methods, however, have restricted capacity in capturing multi-modality, Eg. the VAE-based methods with the diagonal Gaussian prior [1,2,3] or modeling the logit map as a low-rank multivariate normal distribution [4]. This often leads to incorrect estimation of the mode proportions (See Fig. 7 for example), thus resulting in a bias in uncertainty estimation. To address this, we propose an explicit multi-modal uncertainty representation in the form of mixture of partially shared stochastic experts. This design allows us to cluster the segmentation outputs into a set of distinctive styles (i.e., modes) with a better estimation of their proportions (by the gating network).
> > >
> > > While there are potentially different choices of MoSE architecture, we developed a set of partially-shared implicit density networks combined with a light-weight gating network, which provides us three key benefits: 1) it enables us to employ a more effective distribution loss naturally fit with the implicit model (as in WGAN); 2) the partially-shared encoder-decoder architecture allows us to learn the common modes more easily from a shared image representation; and 3) we are able to control the variance of each mode by injecting a simple mode-specific Gaussian noise into the proper layer of a shared decoder network. Those benefits are verified by our experiments and in the end, our method generates an efficient and easy-to-interpret uncertainty representation.
> > >
> > > **Intuition:** Intuitively, our method employs a divide-and-conquer strategy to capture complex multi-modal distributions: each expert network focuses on an individual mode, which is simpler to model, and the gating network estimates the proportions of those learned modes, which leads to a better alignment with the groundtruth. By contrast, the previous methods typically utilize a single network to capture the complex distribution as a whole and suffer from inaccurate estimation (particularly on mode proportions). Moreover, our OT-based distribution loss encourages grouping of segmentation masks based on their similarities (defined by the cost matrix), and hence similar groundtruth annotations can be matched to and learned by the same expert network. Finally, by our design, each expert network is able to capture local variations within a segmentation style while its variance is determined by both our noise injection strategy and model selection (via validation). We note that previous works are unable to exploit such a clustering structure in uncertainty representation.
> > >
> > > [1] Simon Kohl et al. A probabilistic u-net for segmentation of ambiguous images. NeurIPS, 2018.
> > > [2] Simon Kohl et al. A hierarchical probabilistic u-net for modeling multi-scale ambiguities. NeurIPS Workshop, 2019.
> > > [3] Christian F Baumgartner et al. Phiseg: Capturing uncertainty in medical image segmentation. MICCAI, 2019.
> > > [4] Monteiro et al, Stochastic segmentation networks: Modelling spatially correlated aleatoric uncertainty, NeurIPS 2020.

---

> > > ### Author Response · Authors · 2022-12-05
> > > **Gentle reminder for further discussions**
> > >
> > > Dear Reviewer VqFH,
> > >
> > > We deeply appreciate your valuable feedback on improving our paper. We hope our response is able to address your concern about the motivation and intuitions on why our method works well.
> > >
> > > As the deadline for the discussion period is approaching, we would be grateful if you could let us know if there are any further questions or suggestions. We will do our best to respond to them.
> > >
> > > Best regards,
> > > Authors of Paper 1385

---

> > > ### Author Response · Authors · 2022-12-12
> > > **Your feedback is important to us**
> > >
> > > Dear Reviewer VqFH,
> > >
> > > As we are hearing the rebuttal is closing up today, we would be grateful if you could let us know if our response addresses your concerns on the **motivation and intuition on why our method works well**.
> > >
> > > Your feedback is really valuable to us, and we will treat it earnestly.
> > >
> > > Best regards,
> > > Authors of Paper 1385

---

> ### Author Response · Authors · 2022-11-19
> **Response to Reviewer VqFH - Part 4**
>
> **Response to question 4 on relation to ensemble**
>
> As we discussed in the general response, our mixture model is able to learn a clustering of the segmentation process, which differs from the ensemble models. Even if we set the sample number per expert S=1, our OT-based loss will still encourage grouping of similar segmentation outcomes and enable each expert to focus on a group or mode.
>
> To verify this, we perform an additional comparison with S=1 in our model. We first train that model with a standard IoU loss (c.f. Table2 Row1),  which is treated as an ensemble model. We then compare its performance with the model trained with our OT-based loss (with S=1) in Table A2 below. We can see that our method with OT-based loss still achieves better performance than the base ensemble model.
>
> Table A2. Compare our method with ensemble models under S=1. Experiments are conducted on the LIDC dataset with full annotations and evaluated by taking one sample from each expert.
> | Eval          | GED ↓        | M-IOU ↑      | ECE（%）↓    |
> |---------------|--------------|--------------|--------------|
> | S=1, IoU loss | 0.540 ±0.006 | 0.529 ±0.003 | 0.279 ±0.008 |
> | S=1, OT loss  | 0.220 ±0.011 | 0.653 ±0.005 | 0.114 ±0.020 |
>
> **Response to question 5 on annotation and mode**
>
> We want to clarify that we do not assume that each ground truth annotation (or class in Cityscapes) contains exactly one mode. As such, each ground truth annotation may have subgroups forming different modes. In general, the number of experts is determined with prior knowledge or tuned as a hyperparameter. For instance, we can choose the first inflection point around the minimum of the validation curve to obtain a compact model (as shown in Figures 5 and 8 in the Appendix). Moreover, we also found that our model performance is more or less robust under a range of expert numbers.
>
> **Response to question 6 on mode variation**
>
> Similar to the standard mixture models (e.g., GMM), our MoSE uses a data-driven strategy to learn the modes and their variations from the data. Our stochastic experts, defined by a shared encoder-decoder backbone and a set of Gaussian noise priors ($\{m_k,\sigma_k\}$), represent the mode locations and their variations. We agree there are potentially many solutions to such mode decomposition. Our method encourages a compact representation of the modes by controlling the model's hyperparameters  (eg. the number of experts) based on a validation set.
>
> Empirically, we find that the model does capture a reasonable set of modes. Specifically, on the LIDC dataset, we observe three major modes including a mode capturing non-caner annotations, a mode for large cancer annotations, and a mode for small cancer annotations, which is largely consistent with the annotation process [9]. Moreover, in our ablation study,  we find that the model with only two experts performs worse than the four-expert one (see Appendix Fig. 5).

---

> ### Author Response · Authors · 2022-11-19
> **Response to Reviewer VqFH - Part 3**
>
> **Response to question 1 on the expert frequency**
>
> Our mixture model with the OT-based loss performs an implicit clustering of the segmentation masks, which enables the gating network to learn the probability weights of those clusters. Specifically, we note that different annotators can generate similar annotations. For instance, in LIDC dataset, different annotators could agree on no-cancer diagnose, corresponding to the same blank annotations. As such, the coupling matrix P would match these similar annotations to one expert model, leading to a pseudo marginal probability for this expert model larger than uniform probabilities. This further supervises the gating network to learn meaningful mode probability predictions (cf. Eq(6)).
>
> The learned probabilities represent how likely a given input image is annotated in different modes. For example, in the right part of Figure 2, the 1st, 3rd and 4th annotations look largely similar, and they may correspond to the first row of predicted samples captured by expert model 0, and the corresponding probability approximates 3/4.
>
> Moreover, the probability for ground truth annotation $v_n$ can be used as an importance prior in practice. For example, an annotation given by a 10-year labeler has more importance weight than a 1-year annotator. In order to study this more general case,  we have non-uniform $v_n$on the multi-modal cityscapes dataset where annotations have different probabilities. The results show that our method can also perform well under such a setting.
>
> **Response to question 2 on the limit case of many experts:**
>
> We conduct additional experiments with an increasing number of experts in LIDC and Cityscapes. In both cases,  we observe that when the expert number is much larger (eg. 20 experts on LIDC and 50 experts on Cityscapes), the model performance decreases, corresponding to an increase of GED in Figure 5 and Figure 8. One potential reason is that given many experts, it becomes difficult to learn a stable gating network.
>
> In general, we consider the number of experts K as a hyperparameter, which should be determined via model selection (as we did in this work). The above results also demonstrate that the effectiveness of our method does not owe to a large number of experts, which is different from the working mechanism of the ensembling model. We have added those larger expert number points in the revised version of Fig. 5 and Fig.8 and the corresponding discussion in the revised appendix.
>
> **Response to question 3 on a row of ablation study**
>
> We follow the reviewer's suggestion and conduct an additional ablation study with "stochastic experts, uniform expert weights and the IoU-loss". The results are almost the same as our "learnable" case. The reason is that for the setting with the IoU loss in Table 2, we simply adopt an average loss calculated on every possible pair of expert predictions and labels. This average IoU loss provides no supervision to the gating network and hence the expert weights remain the same as initialized (typically uniform). On the other hand, under the average IoU, the experts tend to converge to an average mode, which performs worse than our standard setting due to a limited model capacity. To clarify this, we revise the "learnable" to "learnable/uniform' for expert weight in Table 2, Row 1.
>
> We also add another ablative case in which we can tune the parameters of the gating network under the IoU loss. Specifically, we adopt a weighted IoU-loss where the IoU loss for each prediction is weighted by the corresponding probability. Formally, the loss can be written as $ \ell = \sum_{i,j}\mathbf{u}_n^{(i)}(\theta^{\pi})\text{IoU}(s_n^{(i)}(\theta_s), y_n^{(j)}) $, where $\mathbf{u}_n^{(i)}$, $s_n^{(i)}$, $y_n^{(j)}$ are predicted probability, segmentation output, and ground truth annotation as in Eq.(6). We observe that the weighted IoU loss often leads to a mode collapse where only one expert has a non-zero weight. The results are also shown in Table A1, which is slightly worse than the setting with uniform expert weights due to the single expert result.
>
> Table A1. Compare two ablative cases with our standard setting.
>
> | Expert type | Expert weight       | loss              | GED ↓               | M-IOU ↑           | ECE（%）↓           |
> |-------------|---------------------|-------------------|---------------------|-------------------|---------------------|
> |  stochastic | learnable / uniform | IoU loss          | 0.533 ±0.001        | 0.533 ±0.001      | 0.277 ±0.016        |
> |  stochastic | learnable           | weighted IoU loss | 0.544 ±0.003        | 0.527 ±0.001      | 0.287 ±0.006        |
> |  stochastic | learnable           | OT loss           | **0.218 ± 0.003** | **0.624 ± 0.004** | **0.064 ± 0.015** |

---

> ### Author Response · Authors · 2022-11-19
> **Response to Reviewer VqFH - Part 2**
>
> **Response to weakness 3 on why our method works:**
>
> Thanks for your constructive comment. We first present a general discussion on how our method works here, and leave details of our specific responses to Questions 1-8 (listed below).
>
> The working mechanism of our method can be summarized in two-fold:
> 1) Our mixture of stochastic expert (MoSE) model has a rich capacity to capture multiple modes and to control the variation in each mode (See our response to Q6 below for more details.)
> 2) Our Wasserstein-like loss allows us to directly minimize the distribution distance between the predictive distribution and the ground truth distribution. We further explain how the mode probabilities can be learned via this loss in our response to Q1.
>
> Moreover, we discuss the differences between our model design and the ensemble model in responses to Q2 and Q4, and why our model performs better than the previous methods in response to Q8. All those discussions have been added in Appendix E of the revised paper.
>
> **Response to weakness 4 about the Wasserstein Dice loss in segmentation**
>
> Thanks for pointing out the related work on Wasserstein Dice loss in multi-class segmentation [7,8]. We note that despite Lucas et al.[7,8] also use the Wasserstein distance, our work differs from them significantly in the following three aspects:
>
> 1) Motivation: Lucas et al. [7,8] adopt the Wasserstein distance at the pixel level in which the cost matrix, named 'distance matrix' in [7], is defined among per-pixel classes. This serves as a prior knowledge of inter-class relationships. By comparison, we define Wasserstein distance at the image level, and the cost matrix is calculated among predicted segmentation masks and ground truth masks. Our design measures the distance between distributions of segmentation masks.
>
> 2) Variables: Their optimal transport problem aims to optimize the pixel-wise class probabilities. By contrast, we jointly optimize the segmentation predictions (reflected in the cost matrix) and their probabilities (generated by the expert gating networks).
>
> 3) Objective function: In [7,8], the Wasserstein distance is wrapped as a component in the Dice loss, while we define the Wasserstein distance as the overall objective function. We also develop a constraint relaxation strategy for fast optimization which is not used in [7,8].
>
> We add those references and more discussion for clarification in the related work section of the revised paper.

---

> ### Author Response · Authors · 2022-11-19
> **Response to Reviewer VqFH - Part 1**
>
> **Response to weakness 1 on comparability:**
>
> Thanks for your constructive suggestion. As demonstrated in the general response, our model has comparable parameters to the benchmark models. Also, we keep the unet backbone, data processing procedure, augmentation, etc. same as previous work [2,3] to ensure a fair comparison (cf. Appendix A.1).  We have added more clarification on the comparability in the revised paper.
>
> All previous methods have released their official implementations. We calculate the model size according to their official codes. The scores we used in the comparison tables are all officially reported in previous publications, except where otherwise noted.
>
> **Response to weakness 2 about constraint relaxation:**
>
> In the following, we discuss the theoretical connection of our constraint relaxation to the original formulation. We have also added this discussion in Appendix B of the revised paper.
>
> For clarity, we consider a bi-level optimization formulation in which we adopt an alternating optimization process between $\theta_s$ (outer loop) and $\{P,\theta_{\pi}\}$ (inner loop). We first focus on the inner loop when the parameters for segmentation predictions $\theta_s$ are fixed, and analyze the optimization behavior of $\{P,\theta_{\pi}\}$ in the original form (Eq. (5)) and relaxed form (Eq. (6)).  For simplicity, we rewrite them as following two functions:
>
> $l_1(u,P) = \langle  P , C \rangle \quad u\in \mathbb{A}, P\in\mathbb{B}(u) = \\{P\in \mathbb{R}^{+}: P\mathbf{1}_M = u, P^T\mathbf{1}_N = v\\}$
>
> $l_2(u,P)=  \langle P,C\rangle + \beta KL(P\mathbf{1}_M || u) \quad u\in \mathbb{A}, P\in\mathbb{C} = \\{P\in \mathbb{R}^{+}: P^T\mathbf{1}_N = v\\}$
>
> where we use $C$ short for $C(s(\theta_s),y)$, and $u$ short for $u(\theta_{\pi})$ for notation clarity. We use $\mathbb{A}$ to denote all possible values of $u$ that the gating network can generate, which also satisfies   $u\in \mathbb{R}^+, \sum_i u_i = 1$. The inequality constraint in Eq. (6) is not included in $l_2$ since it will eventually disappear due to annealing. We show in *Proposition 1* that when $\beta$ is sufficiently large, $l_2$ achieves the same optimal solutions as $l_1$ when the KL term is zero. After that, when we alternate to optimizing $\theta_s$ , the two problems are exactly the same as they use the same $P$ from the inner loop. Therefore the overall results of the original problem and the relaxed problem are the same.
>
> *Proposition 1: When $\beta$ is sufficiently large, $l_2$ achieves the same optimal solutions as $l_1$.*
> *Proof:* See Appendix B.
>
> In our optimization algorithm, we take a further step, alternating between $P$ and $u$. Specifically, we first optimize $\bar{P} =$$\arg\min_{P\in \mathbb{C}}\langle P,C\rangle$ and then optimize $\bar{u}$$= \arg\min_{{u}\in\mathbb{A}}KL(\bar P\mathbf{1}_M || u)$ (cf. Eq. (6)). Compared to the joint optimization, this relaxed optimization strategy accelerates the optimization speed but has no guarantee to find the optimal solution as in the original form since the KL term may not achieve zero. Specifically, we demonstrate in *Proposition 2* that when the KL term can be optimized to 0, $\bar{P},\bar{u}$ is still the optimal solution for $l_2$ and $l_1$. Empirically, we find that our model achieves KL divergence around 1e-5, which is much smaller than the scale of the segmentation loss in 1e-1 on the Cityscapes dataset.
>
> *Proposition 2:  In our algorithm, when the KL term can be optimized to 0, $\bar{P} =$$\arg\min_{P\in \mathbb{C}} \langle P,C\rangle$, $\bar{u}$$= \arg\min_{{u}\in\mathbb{A}}KL(\bar P\mathbf{1}_M || u)$ is still the optimal solution for $l_2$ and $l_1$.*
> *Proof:* See Appendix B.

---

### Author Response · Authors · 2022-11-19
**General Response**

We thank all the reviewers for their time and constructive comments. We are pleased that the reviewers recognize the novelty (Reviewers VqFH, LSxs), originality(Reviewer bd5h), and effectiveness(Reviewers VqFH, LSxs, bd5h) of our method and find our writing has a good structure (Reviewer VqFH) and is easy to follow (Reviewer LSxs). In the following, we address some shared concerns in this general response and reply to each individual question by replying to each reviewer. We also make some improvements to our paper according to the suggestions of the reviewers, which are highlighted in blue in the revised version.

**Fairness in comparison with benchmark models (Reviewers VqFH, LSxs):**

We clarify our comparisons by showing the model size of all the benchmark methods in the Table 1 and 3 of the revised paper. We can see that, compared to most benchmark methods, our model has a smaller or comparable number of parameters while achieving the SOTA performance, which demonstrates its efficacy.  In particular, we keep the segmentation backbone in our model the same as in previous work [2,3]. Notably, as shown in Appendix Table 4, our model only adds a small number of parameters (0.33M, 0.8%) to the original Unet backbone (cf. Sec 3.2.1), which is much smaller than the VAE-based methods[1,2].

In addition, in order to keep a fair comparison to the methods using a smaller backbone [4], we conduct experiments with a lighter unet backbone same as in [4] (decrease from the original 6 down-sample layers to 3 down-sample layers), and we find that the performance of our method remains stable (See 'Ours-light' in Table 1, 3), and also achieves the SOTA results in the benchmark. We also add experiments on the Cityscapes dataset with no additional input to keep a fair comparison with ProbUnet [1] (cf. Table 3).

**Compare with the ensembling-based method (Reviewers VqFH, LSxs):**

While our MoSE architecture shares certain similarities with the ensemble-based models (i.e., a combination of simpler models),  there are three key differences between our framework and the ensemble methods:

1) Different goals: Most mixture-based ensemble methods aim to learn a predictor with a reduced model variance by aggregating multiple variants of the base models. By contrast, we use the mixture architecture to explicitly capture the multi-modal distribution caused by the aleatoric uncertainty, in which each component corresponds to a different mode. We intend to learn those modes from data without explicit supervision on the mode identity.

2) Learning strategy:  The ensemble methods typically learn each component model based on the original task loss (mostly with a unique groundtruth for each input), and it is usually unclear which part of the data distribution it captures. In contrast, we design a Wasserstein-like loss to minimize the distribution distance between the model outputs and the groundtruth. Thanks to the OT-based formulation, each expert of our model learns to capture a specific mode in the dataset.

3) Representation: We develop a MoSE structure specifically for our uncertainty modeling task. Our stochastic expert models share most of their parameters and only vary in the latent distributions, which facilitates the learning of compact and meaningful modes. This differs from the conventional design of the component models with totally different parameters in the ensemble methods.

Empirically, we have conducted experiments with our OT-based loss replaced by the IoU-loss (cf. Sec. 4.3.1), which is largely equivalent to learning a standard ensemble model. As shown in Table 2 Row#1, this leads to a performance drop of more than 0.3 in GED, demonstrating that a naive ensembling technique may not suffice for this task. We note that Kohl et al.[1] have also tested the performance of an Unet ensemble method and reached a similar conclusion.

[1] Simon Kohl et al. A probabilistic u-net for segmentation of ambiguous images. NIPS, 2018.
[2] Christian F Baumgartner et al. Phiseg: Capturing uncertainty in medical image segmentation. MICCAI, 2019.
[3] Monteiro et al, Stochastic segmentation networks: Modelling spatially correlated aleatoric uncertainty, NeurIPS 2020.
[4] Shi Hu et al. Supervised uncertainty quantification for segmentation with multiple annotations. MICCAI, 2019.

---

### Decision · Program_Chairs · 2023-01-20

**Decision:**

Accept: poster

**Justification For Why Not Higher Score:**

The reviews do not support it

**Justification For Why Not Lower Score:**

All reviewers suggest acceptance

**Metareview: Summary, Strengths And Weaknesses:**

Summary:
This paper presents a new mixture of stochastic experts method for modeling aleatoric segmentation uncertainty.

Strengths:
- Well written paper with novel ideas on segmentation uncertainty
- Thorough experimental validation

Weaknesses:
- Limited discussion of the model beyond the appendix. Some discussion should be added to the paper itself.
- Missing references
- Some relevant baselines are omitted, also from discussion, e.g. VAE with Gaussian micture prior
- Validation with GED metric has limited ability to draw conclusions

**Note From Pc:**

if the above contains the word "oral" or "spotlight" please see: "oral" presentation means -> notable-top-5% and "spotlight" means -> notable-top-25%. As stated in our emails, we are disassociating presentation type from AC recommendations